# Improved guarantees and a multiple-descent curve for Column Subset Selection and the Nyström method

**Michał Dereziński**
Department of Statistics
University of California, Berkeley
mderezin@berkeley.edu

**Rajiv Khanna**
Department of Statistics
University of California, Berkeley
rajivak@berkeley.edu

**Michael W. Mahoney**
ICSI and Department of Statistics
University of California, Berkeley
mmahoney@stat.berkeley.edu

## Abstract

The Column Subset Selection Problem (CSSP) and the Nyström method are among the leading tools for constructing small low-rank approximations of large datasets in machine learning and scientific computing. A fundamental question in this area is: how well can a data subset of size $k$ compete with the best rank $k$ approximation? We develop techniques which exploit spectral properties of the data matrix to obtain improved approximation guarantees which go beyond the standard worst-case analysis. Our approach leads to significantly better bounds for datasets with known rates of singular value decay, e.g., polynomial or exponential decay. Our analysis also reveals an intriguing phenomenon: the approximation factor as a function of $k$ may exhibit multiple peaks and valleys, which we call a multiple-descent curve. A lower bound we establish shows that this behavior is not an artifact of our analysis, but rather it is an inherent property of the CSSP and Nyström tasks. Finally, using the example of a radial basis function (RBF) kernel, we show that both our improved bounds and the multiple-descent curve can be observed on real datasets simply by varying the RBF parameter.

## 1 Introduction

We consider the task of selecting a small but representative sample of column vectors from a large matrix. Known as the Column Subset Selection Problem (CSSP), this is a well-studied combinatorial optimization task with many applications in machine learning (e.g., feature selection, see Guyon & Elisseeff, 2003; Boutsidis et al., 2008), scientific computing (e.g., Chan & Hansen, 1992; Drineas et al., 2008) and signal processing (e.g., Balzano et al., 2010). In a commonly studied variant of this task, we aim to minimize the squared error of projecting all columns of the matrix onto the subspace spanned by the chosen column subset.

**Definition 1** (CSSP). *Given an $m \times n$ matrix* **A**, *pick a set* $S \subseteq \{1, ..., n\}$ *of $k$ column indices, to minimize*

$$\mathrm{Er}_{\mathbf{A}}(S) := \|\mathbf{A} - \mathbf{P}_S \mathbf{A}\|_F^2,$$

*where* $\| \cdot \|_F$ *is the Frobenius norm,* $\mathbf{P}_S$ *is the projection onto* $\mathrm{span}\{\mathbf{a}_i : i \in S\}$ *and* $\mathbf{a}_i$ *denotes the $i$th column of* **A**.

Another variant of the CSSP emerges in the kernel setting under the name *Nyström method* (Williams & Seeger, 2001; Drineas & Mahoney, 2005; Gittens & Mahoney, 2016). We also discuss this variant,

showing how our analysis applies in this context. Both the CSSP and the Nyström method are ways of constructing accurate low-rank approximations by using submatrices of the target matrix. Therefore, it is natural to ask how close we can get to the best possible rank $k$ approximation error:

$$\text{OPT}_k := \min_{\mathbf{B}:\, \text{rank}(\mathbf{B})=k} \|\mathbf{A} - \mathbf{B}\|_F^2 \leqslant \min_{S:\, |S|=k} \text{Er}_{\mathbf{A}}(S).$$

Our goal is to find a subset $S$ of size $k$ for which the ratio between $\text{Er}_{\mathbf{A}}(S)$ and $\text{OPT}_k$ is small. Furthermore, a brute force search requires iterating over all $\binom{n}{k}$ subsets, which is prohibitively expensive, so we would like to find our subset more efficiently.

Extensive literature has been dedicated to developing algorithms for the CSSP (an in depth discussion of the related work can be found in Appendix A). In terms of worst-case analysis, Deshpande et al. (2006) gave a randomized method which returns a set $S$ of size $k$ such that:

$$\frac{\mathbb{E}[\text{Er}_{\mathbf{A}}(S)]}{\text{OPT}_k} \leqslant k + 1. \tag{1}$$

While the original algorithm was slow, efficient implementations have been provided since then (e.g., see Deshpande & Rademacher, 2010; Dereziński, 2019). The method belongs to the family of cardinality constrained Determinantal Point Processes (DPPs, see Dereziński & Mahoney, 2020), and will be denoted as $S \sim k\text{-DPP}(\mathbf{A}^\top\mathbf{A})$. The approximation factor $k + 1$ is optimal in the worst-case, since for any $0 < k < n \leqslant m$ and $0 < \delta < 1$, an $m \times n$ matrix $\mathbf{A}$ can be constructed for which $\frac{\text{Er}_{\mathbf{A}}(S)}{\text{OPT}_k} \geqslant (1-\delta)(k+1)$ for all subsets $S$ of size $k$. Yet it is known that, in practice, CSSP algorithms perform better than worst-case, so the question we consider is: how can we go beyond the usual worst-case analysis to accurately reflect what is possible in the CSSP?

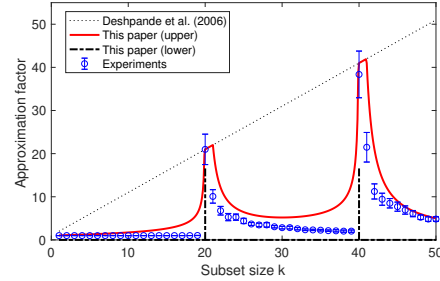

Figure 1: Empirical study of the expected approximation factor $\mathbb{E}[\text{Er}_{\mathbf{A}}(S)]/\text{OPT}_k$ for a $k$-DPP with different subset sizes $|S| = k$, compared to our theory. We use a data matrix $\mathbf{A}$ whose spectrum exhibits two sharp drops, demonstrating multiple-descent. The lower bounds are based on Theorem 3, whereas, as our upper bound, we plot the minimum over all $\Phi_s(k)$ from Theorem 1. Note that multiple-descent vanishes under smooth spectral decay, resulting in improved guarantees (see Theorem 2 and Figure 2).

**Contributions.** We provide improved guarantees for the CSSP approximation factor, which go beyond the worst-case analysis and which lead to surprising conclusions.

1. New upper bounds: We develop a family of upper bounds on the CSSP approximation factor (Theorem 1), which we call the Master Theorem as they can be used to derive a number of new guarantees. In particular, we show that when the data matrix $\mathbf{A}$ exhibits a known spectral decay, then (1) can often be drastically improved (Theorem 2).

2. New lower bound: Even though the worst-case upper bound in (1) can often be loose, there are cases when it cannot be improved. We give a new lower bound construction (Theorem 3) showing that there are matrices $\mathbf{A}$ for which multiple different subset sizes exhibit worst-case behavior.

3. Multiple-descent curve: Our upper and lower bounds reveal that for some matrices the CSSP approximation factor can exhibit peaks and valleys as a function of the subset size $k$ (see Figure 1). We show that this phenomenon is an inherent property of the CSSP (Corollary 1).

## 1.1 Main results

Our upper bounds rely on the notion of effective dimensionality called stable rank (Alaoui & Mahoney, 2015). Here, we use an extended version of this concept, as defined by Bartlett et al. (2019).

**Definition 2** (Stable rank). *Let $\lambda_1 \geqslant \lambda_2 \geqslant \dots$ denote the eigenvalues of the matrix $\mathbf{A}^\top\mathbf{A}$. For $0 \leqslant s < \text{rank}(\mathbf{A})$, we define the stable rank of order $s$ as $\text{sr}_s(\mathbf{A}) = \lambda_{s+1}^{-1} \sum_{i>s} \lambda_i$.*

In the following result, we define a family of functions $\Phi_s(k)$ which bound the approximation factor $\text{Er}_{\mathbf{A}}(S)/\text{OPT}_k$ in the range of $k$ between $s$ and $s + \text{sr}_s(\mathbf{A})$. We call this the Master Theorem because we use it to derive a number of more specific upper bounds.

**Theorem 1** (Master Theorem). *Given* $0 \leqslant s < \mathrm{rank}(\mathbf{A})$, *let* $t_s = s + \mathrm{sr}_s(\mathbf{A})$, *and suppose that* $s + \frac{7}{\epsilon^4} \ln^2 \frac{1}{\epsilon} \leqslant k \leqslant t_s - 1$, *where* $0 < \epsilon \leqslant \frac{1}{2}$. *If* $S \sim k\text{-DPP}(\mathbf{A}^\top \mathbf{A})$, *then*

$$\frac{\mathbb{E}[\mathrm{Er}_\mathbf{A}(S)]}{\mathrm{OPT}_k} \leqslant (1 + 2\epsilon)^2\, \Phi_s(k), \qquad where \quad \Phi_s(k) = \left(1 + \tfrac{s}{k-s}\right)\sqrt{1 + \tfrac{2(k-s)}{t_s - k}}\,.$$

Note that we separated out the dependence on $\epsilon$ from the function $\Phi_s(k)$, because the term $(1 + 2\epsilon)^2$ is an artifact of a concentration of measure analysis that is unlikely to be of practical significance. In fact, we believe that the dependence on $\epsilon$ can be eliminated from the statement entirely (see Conjecture 1).

We next examine the consequences of the Master Theorem, starting with a sharp transition that occurs as $k$ approaches the stable rank of $\mathbf{A}$.

**Remark 1** (Sharp transition). *For any $k$ it is true that:*

1. *For all $\mathbf{A}$, if $k \leqslant \mathrm{sr}_0(\mathbf{A}) - 1$, then there is a subset $S$ of size $k$ such that $\frac{\mathrm{Er}_\mathbf{A}(S)}{\mathrm{OPT}_k} = O(\sqrt{k}\,)$.*

2. *There is $\mathbf{A}$ such that $\mathrm{sr}_0(\mathbf{A}) - 1 < k < \mathrm{sr}_0(\mathbf{A})$ and for every size $k$ subset $S$, $\frac{\mathrm{Er}_\mathbf{A}(S)}{\mathrm{OPT}_k} \geqslant 0.9\,k$.*

Part 1 of the remark follows from the Master Theorem by setting $s = 0$, whereas part 2 follows from the lower bound of Guruswami & Sinop (2012). Observe how the worst-case approximation factor jumps from $O(\sqrt{k}\,)$ to $\Omega(k)$, as $k$ approaches $\mathrm{sr}_0(\mathbf{A})$. An example of this sharp transition is shown in Figure 1, where the stable rank of $\mathbf{A}$ is around 20.

While certain matrices directly exhibit the sharp transition from Remark 1, many do not. In particular, for matrices with a known rate of spectral decay, the Master Theorem can be used to provide improved guarantees on the CSSP approximation factor over *all* subset sizes.

To illustrate this, we give novel bounds for the two most commonly studied decay rates: polynomial and exponential.

**Theorem 2** (Examples without sharp transition). *Let $\lambda_1 \geqslant \lambda_2 \geqslant \ldots$ be the eigenvalues of $\mathbf{A}^\top \mathbf{A}$. There is an absolute constant $c$ such that for any $0 < c_1 \leqslant c_2$, with $\gamma = c_2/c_1$, if:*

1. *(**polynomial spectral decay**) $c_1 i^{-p} \leqslant \lambda_i \leqslant c_2 i^{-p}\ \forall_i$, with $p > 1$, then $S \sim k\text{-DPP}(\mathbf{A}^\top \mathbf{A})$ satisfies*

$$\frac{\mathbb{E}[\mathrm{Er}_\mathbf{A}(S)]}{\mathrm{OPT}_k} \leqslant c\gamma p.$$

2. *(**exponential spectral decay**) $c_1(1-\delta)^i \leqslant \lambda_i \leqslant c_2(1-\delta)^i\ \forall_i$, $\delta \in (0,1)$, then $S \sim k\text{-DPP}(\mathbf{A}^\top \mathbf{A})$ satisfies*

$$\frac{\mathbb{E}[\mathrm{Er}_\mathbf{A}(S)]}{\mathrm{OPT}_k} \leqslant c\gamma(1 + \delta k).$$

Note that for polynomial decay, unlike in (1), the approximation factor is constant, i.e., it does not depend on $k$. For exponential decay, our bound provides an improvement over (1) when $\delta = o(1)$. To illustrate how these types of bounds can be obtained from the Master Theorem, consider the function $\Phi_s(k)$ for some $s > 0$. The first term in the function, $1 + \frac{s}{k-s}$, decreases with $k$, whereas the second term (the square root) increases, albeit at a slower rate. This creates a U-shaped curve which, if sufficiently wide, has a valley where the approximation factor can get arbitrarily close to 1. This will occur when $\mathrm{sr}_s(\mathbf{A})$ is large, i.e., when the spectrum of $\mathbf{A}^\top \mathbf{A}$ has a relatively flat region after the $s$th eigenvalue (Figure 1 for $k$ between 20 and 40). Note that a peak value of some function $\Phi_{s_1}$ may coincide with a valley of some $\Phi_{s_2}$, so only taking a minimum over all functions reveals the true approximation landscape predicted by the Master Theorem. To prove Theorem 2, we show that the stable ranks $\mathrm{sr}_s(\mathbf{A})$ are sufficiently large so that any $k$ lies in the valley of some function $\Phi_s(k)$ (see Section 2).

The peaks and valleys of the CSSP approximation factor suggested by Theorem 1 are in fact an inherent property of the problem, rather than an artifact of our analysis or the result of using a particular algorithm. We prove this by constructing a family of matrices $\mathbf{A}$ for which the best possible approximation factor is large, i.e., close to the worst-case upper bound of Deshpande et al. (2006), not just for one size $k$, but for a sequence of increasing sizes.

**Theorem 3** (Lower bound). *For any $\delta \in (0,1)$ and $0 = k_0 < k_1 < ... < k_t < n \leqslant m$, there is a matrix $\mathbf{A} \in \mathbb{R}^{m \times n}$ such that for any subset $S$ of size $k_i$, where $i \in \{1, ..., t\}$,*

$$\frac{\text{Er}_{\mathbf{A}}(S)}{\text{OPT}_{k_i}} \geqslant (1 - \delta)(k_i - k_{i-1}).$$

Combining the Master Theorem with the lower bound of Theorem 3 we can easily provide an example matrix for which the optimal solution to the CSSP problem exhibits multiple peaks and valleys. We refer to this phenomenon as the multiple-descent curve.

**Corollary 1** (Multiple-descent curve). *For $t \in \mathbb{N}$ and $\delta \in (0,1)$, there is a sequence $0 < k_1^l < k_1^u < k_2^l < k_2^u < ... < k_t^l < k_t^u$ and $\mathbf{A} \in \mathbb{R}^{m \times n}$ such that for any $i \in \{1, ..., t\}$:*

$$\min_{S:|S|=k_i^l} \frac{\text{Er}_{\mathbf{A}}(S)}{\text{OPT}_{k_i^l}} \leqslant 1 + \delta \qquad and \qquad \min_{S:|S|=k_i^u} \frac{\text{Er}_{\mathbf{A}}(S)}{\text{OPT}_{k_i^u}} \geqslant (1-\delta)(k_i^u + 1).$$

**Connection to double descent.** A number of phase transitions have been recently observed in the machine learning literature which are commonly dubbed *double descent*. The term was introduced by Belkin et al. (2019a) in the context of generalization error of statistical learning, however Poggio et al. (2019) observed that the behavior of the generalization error, at least for linear models, can be explained by a more fundamental double descent phenomenon observed in the condition number of random matrices. Also, Liao et al. (2020) showed that double descent is merely a manifestation of the spectral phase transitions in high-dimensional random kernel matrices. Further, Liang et al. (2020) observed that the spectral phase transitions of certain kernels can lead to multiple double descent peaks. The multiple-descent phenomenon in the CSSP that emerges from our analysis is related to those works in that the spectral properties of the data matrix (and, in particular, the condition number) determine the peaks (i.e., phase transitions) discussed in Corollary 1. Those parallels manifest themselves most clearly when comparing our analysis to the work of Bartlett et al. (2019), which uses the same notion of stable rank as we do, and Dereziński et al. (2019b), where determinantal sampling plays a central role in the analysis of double descent. However, we stress that there are also important differences in our setting: (1) the CSSP is a *deterministic* optimization task, and (2) we study the *approximation factor*, rather than the generalization error (further discussion in Appendix A).

## 1.2 The Nyström method

We briefly discuss how our results translate to guarantees for the Nyström mehod, a variant of the CSSP in the kernel setting which has gained considerable interest in the machine learning literature (Drineas & Mahoney, 2005; Gittens & Mahoney, 2016). In this context, rather than being given the column vectors explicitly, we consider the $n \times n$ matrix $\mathbf{K}$ whose entry $(i,j)$ is the dot product between the $i$th and $j$th vector in the kernel space, $\langle \mathbf{a}_i, \mathbf{a}_j \rangle_{\mathbf{K}}$. A Nyström approximation of $\mathbf{K}$ based on subset $S$ is defined as $\widehat{\mathbf{K}}(S) = \mathbf{C}\mathbf{B}^{\dagger}\mathbf{C}^{\top}$, where $\mathbf{B}$ is the $|S| \times |S|$ submatrix of $\mathbf{K}$ indexed by $S$, whereas $\mathbf{C}$ is the $n \times |S|$ submatrix with columns indexed by $S$. The Nyström method has many applications in machine learning, including for kernel machines (Williams & Seeger, 2001), Gaussian Process regression (Burt et al., 2019) and Independent Component Analysis (Bach & Jordan, 2003).

**Remark 2.** *If $\mathbf{K} = \mathbf{A}^{\top}\mathbf{A}$ and $\| \cdot \|_*$ is the trace norm, then $\left\| \mathbf{K} - \widehat{\mathbf{K}}(S) \right\|_* = \text{Er}_{\mathbf{A}}(S)$ for all $S \subseteq \{1, ..., n\}$. Moreover, the trace norm error of the best rank $k$ approximation of $\mathbf{K}$, is equal to the squared Frobenius norm error of the best rank $k$ approximation of $\mathbf{A}$, i.e.,*

$$\min_{\widehat{\mathbf{K}}: \text{rank}(\mathbf{K})=k} \|\mathbf{K} - \widehat{\mathbf{K}}\|_* = \text{OPT}_k.$$

This connection was used by Belabbas & Wolfe (2009) to adapt the $k + 1$ approximation factor bound of Deshpande et al. (2006) to the Nyström method. Similarly, all of our results for the CSSP, including the multiple-descent curve that we have observed, can be translated into analogous statements for the trace norm approximation error in the Nyström method. Of particular interest are the improved bounds for kernel matrices with known eigenvalue decay rates. Such matrices arise naturally in machine learning when using standard kernel functions such as the Gaussian Radial Basis Function (RBF) kernel (a.k.a. the squared exponential kernel) and the Matérn kernel (Burt et al., 2019).

RBF kernel: If $\langle \mathbf{a}_i, \mathbf{a}_j \rangle_{\mathbf{K}} = \exp(-\|\mathbf{a}_i - \mathbf{a}_j\|^2/\sigma^2)$ and the data comes from $\mathcal{N}(0, \eta^2)$, then, for large $n$, $\lambda_i \asymp \lambda_1(\frac{b}{a+b+c})^i$, where $a = \frac{1}{4\eta^2}$, $b = \frac{1}{\sigma^2}$ and $c = \sqrt{a^2 + 2ab}$ (Santa et al., 1997), so Theorem 2

yields an approximation factor of $O(1 + \frac{a+c}{a+b+c}k)$, better than $k+1$ when $\sigma^2 \ll \eta^2$. Note that the parameter $\sigma$ defines the size of a neighborhood around which the data points are deemed similar by the RBF kernel. Therefore, smaller $\sigma$ means that each data point has fewer similar neighbors.

Matérn kernel: If $\mathbf{K}$ is the Matérn kernel with parameters $\nu$ and $\ell$ and the data is distributed according to a uniform measure in one dimension, then $\lambda_i \asymp \lambda_1 i^{-2\nu-1}$ (Rasmussen & Williams, 2006), so Theorem 2 yields a Nyström approximation factor of $O(1 + \nu)$ for any subset size $k$.

In Section 4, we also empirically demonstrate our improved guarantees and the multiple-descent curve for the Nyström method with the RBF kernel.

## 2 Upper bounds

In this section, we derive the upper bound given in Theorem 1 by using a novel expectation formula for the squared projection error of a DPP. We then show how this result can be used to obtain improved guarantees for matrices with known eigenvalue decays, i.e., Theorem 2. Our analysis heavily relies on the theory of DPPs (Dereziński & Mahoney, 2020), so for completeness, in Appendix B we provide a brief summary of DPPs and the relevant results.

Let $S \sim \mathrm{DPP}(\frac{1}{\alpha}\mathbf{A}^\top\mathbf{A})$ denote a distribution over all subsets $S \subseteq [n]$ so that $\Pr(S) \propto \det(\frac{1}{\alpha}\mathbf{A}_S^\top\mathbf{A}_S)$, where $\alpha > 0$. Then, $k$-$\mathrm{DPP}(\mathbf{A}^\top\mathbf{A})$ is simply a restriction of $\mathrm{DPP}(\frac{1}{\alpha}\mathbf{A}^\top\mathbf{A})$ to the subsets of size $k$ (regardless of the choice of $\alpha$). However, the expected subset size for $\mathrm{DPP}(\frac{1}{\alpha}\mathbf{A}^\top\mathbf{A})$ does depend on $\alpha$. Our analysis relies on a careful selection of this parameter. In Lemma 6 (Appendix B), we show the following expectation formula for the CSSP approximation error:

$$\mathbb{E}[\mathrm{Er}_{\mathbf{A}}(S)] = \mathbb{E}[|S|] \cdot \alpha, \quad \text{for} \quad S \sim \mathrm{DPP}(\tfrac{1}{\alpha}\mathbf{A}^\top\mathbf{A}).$$

If we set $\alpha = \mathrm{OPT}_k = \sum_{i=k+1}^n \lambda_i$, where $\lambda_i$ are the eigenvalues of $\mathbf{A}^\top\mathbf{A}$ in decreasing order, then:

$$\mathbb{E}[|S|] = \sum_{i=1}^n \frac{\lambda_i}{\alpha + \lambda_i} \leqslant \sum_{i=1}^k \frac{\lambda_i}{\alpha + \lambda_i} + 1 \leqslant k + 1.$$

This recovers the upper bound of Deshpande et al. (2006), i.e., $\mathbb{E}[\mathrm{Er}_{\mathbf{A}}(S)] \leqslant (k+1)\mathrm{OPT}_k$, except that the subset size is randomized with expectation bounded by $k+1$, instead of a fixed subset size equal $k$. However, a more refined choice of the parameter $\alpha$ allows us to significantly improve on the above error bound in certain regimes, as shown below.

**Lemma 1.** *For any* $\mathbf{A}$, $0 \leqslant \epsilon < 1$ *and* $s < k < t_s$, *where* $t_s = s + \mathrm{sr}_s(\mathbf{A})$, *say* $S \sim \mathrm{DPP}(\frac{1}{\alpha}\mathbf{A}^\top\mathbf{A})$ *for* $\alpha = \frac{\gamma_s(k)\mathrm{OPT}_k}{(1-\epsilon)(k-s)}$ *and* $\gamma_s(k) := \sqrt{1 + \frac{2(k-s)}{t_s-k}}$. *Then, defining* $\Phi_s(k) := \left(1 + \frac{s}{k-s}\right)\gamma_s(k)$,

$$\frac{\mathbb{E}[\mathrm{Er}_{\mathbf{A}}(S)]}{\mathrm{OPT}_k} \leqslant \frac{\Phi_s(k)}{1-\epsilon} \quad \text{and} \quad \mathbb{E}[|S|] \leqslant k - \epsilon\,\frac{k-s}{\gamma_s(k)}.$$

Note that, setting $\epsilon = 0$, the above lemma implies that we can achieve approximation factor $\Phi_s(k)$ with a DPP whose expected size is bounded by $k$. We introduce $\epsilon$ so that we can convert the bound from DPP to the fixed size k-DPP via a concentration argument. Intuitively, our strategy is to show that the randomized subset size of a DPP is sufficiently concentrated around its expectation that with high probability it will be bounded by $k$, and for this we need the expectation to be strictly below $k$. A careful application of the Chernoff bound for a Poisson binomial random variable yields the following concentration bound.

**Lemma 2.** *Let* $S$ *be as in Lemma 1 with* $\epsilon \leqslant \frac{1}{2}$. *If* $s + \frac{7}{\epsilon^4}\ln^2\frac{1}{\epsilon} \leqslant k \leqslant t_s - 1$, *then* $\Pr(|S| > k) \leqslant \epsilon$.

Finally, any expected bound for random size DPPs can be converted to an expected bound for a fixed size k-DPP via the following result.

**Lemma 3.** *For any* $\mathbf{A} \in \mathbb{R}^{m \times n}$, $k \in [n]$ *and* $\alpha > 0$, *if* $S \sim \mathrm{DPP}(\frac{1}{\alpha}\mathbf{A}^\top\mathbf{A})$ *and* $S' \sim k\text{-}\mathrm{DPP}(\mathbf{A}^\top\mathbf{A})$, *then*

$$\mathbb{E}[\mathrm{Er}_{\mathbf{A}}(S')] \leqslant \mathbb{E}[\mathrm{Er}_{\mathbf{A}}(S) \mid |S| \leqslant k].$$

The above inequality may seem intuitively obvious since adding more columns to a set $S$ to complete it to size $k$ always reduces the error. However, a priori, it could happen that going from subsets of

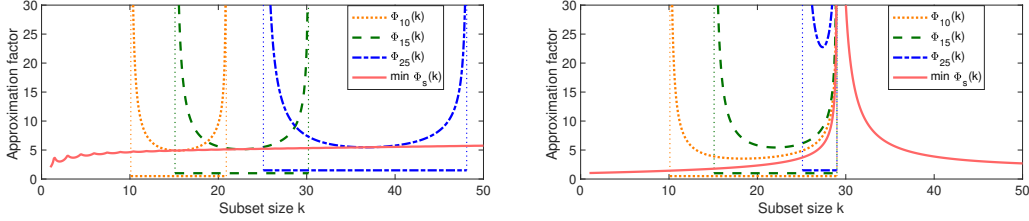

Figure 2: Illustration of the upper bound functions $\Phi_s(k)$ for different values of $s$, with a $200 \times 200$ matrix $\mathbf{A}$ such that the $i$th eigenvalue of $\mathbf{A}^\top \mathbf{A}$ is set to: (left) $1/i$; (right) 1 for $i < 30$ and 0.01 for $i \geqslant 30$. For each function, we marked the window of applicable $k$'s with a horizontal line. For polynomial spectral decay (left), the stable rank $\mathrm{sr}_s(\mathbf{A})$ (i.e., the width of the window starting at $s$) increases, while for the sharp spectrum drop (right) the stable rank shrinks as the window approaches the drop, causing a peak in the upper bound.

size $k-1$ to subsets of size $k$ results in a redistribution of probabilities to the subsets with larger error. To show that this will not happen, our proof relies on classic but non-trivial combinatorial bounds called Newton's inequalities. Putting together Lemmas 1, 2 and 3, we obtain our Master Theorem.

**Proof of Theorem 1** Let $S \sim \mathrm{DPP}(\frac{1}{\alpha}\mathbf{A}^\top\mathbf{A})$ be sampled as in Lemma 1, and let $S' \sim k$-$\mathrm{DPP}(\mathbf{A}^\top\mathbf{A})$. We have:

$$\mathbb{E}\big[\mathrm{Er}_\mathbf{A}(S')\big] \overset{(a)}{\leqslant} \mathbb{E}\big[\mathrm{Er}_\mathbf{A}(S) \mid |S| \leqslant k\big] \leqslant \frac{\mathbb{E}\big[\mathrm{Er}_\mathbf{A}(S)\big]}{\Pr(|S| \leqslant k)} \overset{(b)}{\leqslant} \frac{\Phi_s(k)}{(1-\epsilon)^2} \cdot \mathrm{OPT}_k,$$

where $(a)$ follows from Lemma 3 and $(b)$ follows from Lemmas 1 and 2. Since $0 < \epsilon \leqslant \frac{1}{2}$, we have $\frac{1}{(1-\epsilon)^2} \leqslant (1+2\epsilon)^2$, which completes the proof. ∎

We now demonstrate how Theorem 1 can be used as the Master Theorem to derive new bounds on the CSSP approximation factor under additional assumptions on the singular value decay of matrix $\mathbf{A}$. Rather than a single upper bound, Theorem 1 provides a family of upper bounds $\Phi_s$, each with a range of applicable values $k$. Since each $\Phi_s(k)$ forms a U-shaped curve, its smallest point falls near the middle of that range. In Figure 2 we visualize these bounds as a sliding window that sweeps across the axis representing possible subset sizes. The width of the window varies: when it starts at $s$ then its width is the stable rank $\mathrm{sr}_s(\mathbf{A})$. The wider the window, the lower is the valley of the corresponding U-curve. Thus, when bounding the approximation factor for a given $k$, we should choose the widest window such that $k$ falls near the bottom of its U-curve. Showing a guarantee that holds for all $k$ requires lower-bounding the stable ranks $\mathrm{sr}_s(\mathbf{A})$ for each $s$. This is straightforward for both polynomial and exponential decay. Specifically, using the notation from Theorem 2, in Appendix E we prove that:

$$\mathrm{sr}_s(\mathbf{A}) = \begin{cases} \Omega(s/p), & \text{for polynomial rate } \lambda_i \asymp 1/i^p, \\ \Omega(1/\delta), & \text{for exponential rate } \lambda_i \asymp (1-\delta)^i. \end{cases}$$

As an example, Figure 2 (left) shows that the stable rank $\mathrm{sr}_s(\mathbf{A})$, i.e., the width of the window starting at $s$, grows linearly with $s$ for eigenvalues decaying polynomially with $p = 1$. As a result, the bottom of each U-shaped curve remains at roughly the same level, making the CSSP approximation factor independent of $k$, as in Theorem 2. In contrast, Figure 2 (right) provides the same plot for a different matrix $\mathbf{A}$ with a sharp drop in the spectrum. The U-shaped curves cannot slide smoothly across that drop because of the shrinking stable ranks, which results in a peak similar to the ones observed in Figure 1.

## 3  Lower bound

As discussed in the previous section, our upper bounds for the CSSP approximation factor exhibit a peak (a high point, with the bound decreasing on either side) around a subset size $k$ when there is a sharp drop in the spectrum of $\mathbf{A}$ around the $k$th singular value. It is natural to ask whether this peak is an artifact of our analysis, or a property of the k-DPP distribution, or whether even optimal CSSP subsets exhibit this phenomenon. In this section, we extend a lower bound construction of Deshpande

et al. (2006) and use it to show that for certain matrices the approximation factor of the optimal CSSP subset, i.e., $\min_{|S|=k} \mathrm{Er}_\mathbf{A}(S)/\mathrm{OPT}_k$, can exhibit not just one but any number of peaks as a function of $k$, showing that the multiple-descent curve in Figure 1 describes a real phenomenon in the CSSP.

The lower bound construction of Deshpande et al. (2006) relies on arranging the column vectors of a $(k+1) \times (k+1)$ matrix $\mathbf{A}$ into a centered symmetric $k$-dimensional simplex. This way, the $k+1$ columns are spanning a $k$ dimensional subspace which contains the $k$ leading singular vectors of $\mathbf{A}$. They then proceed to shift the columns slightly in the direction orthogonal to that subspace so that the $(k+1)$st singular value of $\mathbf{A}$ becomes non-zero. This results in an instance of the CSSP with a sharp drop in the spectrum. Due to the symmetry in this construction, all subsets of size $k$ have an identical squared projection error. It is easy to show that this error satisfies $\mathrm{Er}_\mathbf{A}(S) \geqslant (1-\delta)(k+1)\mathrm{OPT}_k$, where $\delta$ is a parameter which depends on the condition number of matrix $\mathbf{A}$ and it can be driven arbitrarily close to 0. Another variant of this construction was also provided by Guruswami & Sinop (2012). The key limitation of both of these constructions is that they only provide a lower bound for a single subset size $k$ in a given matrix, whereas our goal is to show that the CSSP can exhibit the multiple-descent curve, which requires lower bounds for multiple different values of $k$ holding with respect to the same matrix $\mathbf{A}$.

Our strategy for constructing the lower bound matrix is to concatenate together multiple sets of columns, each of which represents a simplex spanning some subspace of $\mathbb{R}^m$. The key challenge that we face in this approach is that, unlike in the construction of Deshpande et al. (2006), different subsets of the same size will have different projection errors.

**Lemma 4.** *Fix $\delta \in (0,1)$ and consider unit vectors $\mathbf{a}_{i,j} \in \mathbb{R}^m$ in general position, where $i \in [t]$, $j \in [l_i]$, such that $\sum_j \mathbf{a}_{i,j} = 0$ for each $i$, and for any $i,j,i',j'$, if $i \neq i'$ then $\mathbf{a}_{i,j}$ is orthogonal to $\mathbf{a}_{i',j'}$. Also, let unit vectors $\{\mathbf{v}_i\}_{i\in[t]}$ be orthogonal to each other and to all $\mathbf{a}_{i,j}$. There are positive scalars $\alpha_i, \beta_i$ for $i \in [t]$ such that matrix $\mathbf{A}$ with columns $\alpha_i \mathbf{a}_{i,j} + \beta_i \mathbf{v}_i$ over all $i$ and $j$ satisfies:*

$$\min_{|S|=k_i} \frac{\mathrm{Er}_\mathbf{A}(S)}{\mathrm{OPT}_{k_i}} \geqslant (1-\delta)l_i, \quad \text{for } k_i = l_1 + \ldots + l_i - 1.$$

**Proof of Theorem 3** We let $l_1 = k_1 + 1$ and then for $i > 1$ we set $l_i = k_i - k_{i-1}$. We then construct the vectors $\mathbf{a}_{i,j}$ that satisfy Lemma 4 by letting each set $\{\mathbf{a}_{i,j}\}_j$ be the corners of a centered $(l_i - 1)$-dimensional regular simplex. We ensure that each simplex is orthogonal to every other simplex by placing them in orthogonal subspaces. ∎

## 4 Empirical evaluation

In this section, we provide an empirical evaluation designed to demonstrate how our improved guarantees for the CSSP and Nyström method, as well as the multiple-descent phenomenon, can be easily observed on real datasets. We use a standard experimental setup for data subset selection using the Nyström method (Gittens & Mahoney, 2016), where an $n \times n$ kernel matrix $\mathbf{K}$ for a dataset of size $n$ is defined so that the entry $(i,j)$ is computed using the Gaussian Radial Basis Function (RBF) kernel: $\langle \mathbf{a}_i, \mathbf{a}_j \rangle_\mathrm{K} = \exp(-\|\mathbf{a}_i - \mathbf{a}_j\|^2/\sigma^2)$, where $\sigma$ is a free parameter. We are particularly interested in the effect of varying $\sigma$. Nyström subset selection is performed using $S \sim k\text{-DPP}(\mathbf{K})$ (Definition 3), and we plot the expected approximation factor $\mathbb{E}[\|\mathbf{K} - \widehat{\mathbf{K}}(S)\|_*]/\mathrm{OPT}_k$ (averaged over 1000 runs), where $\widehat{\mathbf{K}}(S)$ is the Nyström approximation of $\mathbf{K}$ based on the subset $S$ (see Section 1.2), $\|\cdot\|_*$ is the trace norm, and $\mathrm{OPT}_k$ is the trace norm error of the best rank $k$ approximation. Additional experiments, using greedy selection instead of a k-DPP, are in Appendix H. As discussed in Section 1.2, this task is equivalent to the CSSP task defined on the matrix $\mathbf{A}$ such that $\mathbf{K} = \mathbf{A}^\top \mathbf{A}$.

The aim of our empirical evaluation is to verify the following two claims motivated by our theory (and to illustrate that doing so is as easy as varying the RBF parameter $\sigma$):

1. When the spectral decay is sufficiently slow/smooth, the approximation factor for CSSP/Nyström is much better than suggested by previous worst-case bounds.

2. A drop in spectrum around the $k$th eigenvalue results in a peak in the approximation factor near subset size $k$. Several drops result in the multiple-descent curve.

In Figure 3 (top), we plot the approximation factor against the subset size $k$ (in the range of 1 to 40) for an artificial toy dataset and for two benchmark regression datasets from the Libsvm repository

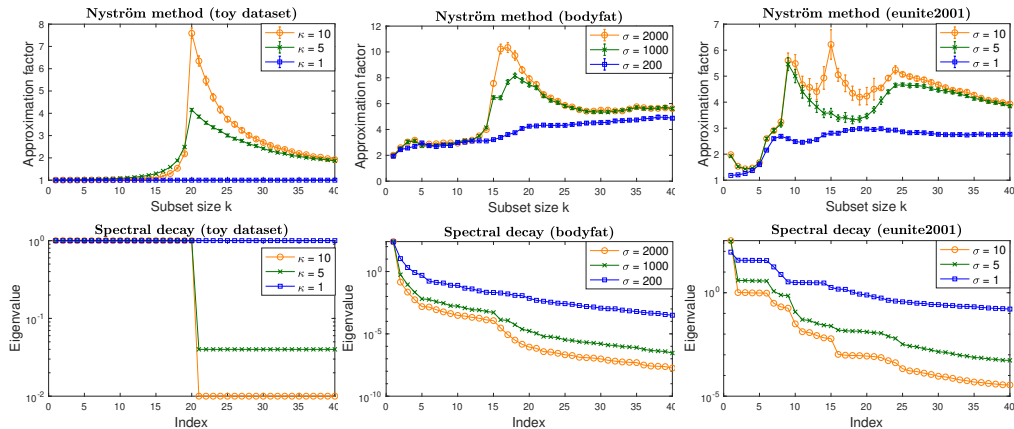

Figure 3: Top three plots show the Nyström approximation factor $\mathbb{E}[\|\mathbf{K} - \widehat{\mathbf{K}}(S)\|_*]/\mathrm{OPT}_k$, where $S \sim k\text{-DPP}(\mathbf{K})$ (experiments using greedy selection instead of a k-DPP are in Appendix H), for a toy dataset ($\kappa$ is the condition number) and two Libsvm datasets ($\sigma$ is the RBF parameter). Error bars show three times the standard error of the mean over 1000 trials. Bottom three plots show the spectral decay for the top 40 eigenvalues of each kernel $\mathbf{K}$. Note that the peaks in the approximation factor align with the drops in the spectrum.

(*bodyfat* and *eunite2001*, see Chang & Lin, 2011). The toy dataset is constructed by scaling the eigenvalues of a random $50 \times 50$ Gaussian matrix so that the spectrum is flat with a single drop at the 21-st eigenvalue. For each dataset, in Figure 3 (bottom), we also show the top 40 eigenvalues of the kernel $\mathbf{K}$ in decreasing order. For the toy dataset, to maintain full control over the spectrum we use the linear kernel $\langle \mathbf{a}_i, \mathbf{a}_j \rangle_K = \mathbf{a}_i^\top \mathbf{a}_j$, and we show results for three different values of the condition number $\kappa$ of kernel $\mathbf{K}$. For the benchmark datasets, we show results on the RBF kernel with three different values of the parameter $\sigma$.

Examining the toy dataset (Figure 3, left), it is apparent that a larger drop in spectrum leads to a sharper peak in the approximation factor as a function of the subset size $k$, whereas a flat spectrum results in the approximation factor being close to 1. A similar trend is observed for dataset *bodyfat* (Figure 3, center), where large parameter $\sigma$ results in a peak that is aligned with a spectrum drop, while decreasing $\sigma$ makes the spectrum flatter and the factor closer to 1. Finally, dataset *eunite2001* (Figure 3, right) exhibits a full multiple-descent curve with up to three peaks for large values of $\sigma$, and the peaks are once again aligned with the spectrum drops. Decreasing $\sigma$ gradually eliminates the peaks, resulting in a uniformly small approximation factor. Thus, both of our theoretical claims can easily be verified on this dataset simply by adjusting the RBF parameter.

While the right choice of the parameter $\sigma$ ultimately depends on the downstream machine learning task, it has been observed that varying $\sigma$ has a pronounced effect on the spectral properties of the kernel matrix, (see, e.g., Gittens & Mahoney, 2016; Lawlor et al., 2016; Wang et al., 2019). The main takeaway from our results here is that, depending on the structure of the problem, we may end up in the regime where the Nyström approximation factor exhibits a multiple-descent curve (e.g., due to a hierarchical nature of the data) or in the regime where it is relatively flat.

## 5   Conclusions and open problems

We derived new guarantees for the Column Subset Selection Problem (CSSP) and the Nyström method, going beyond worst-case analysis by exploiting the structural properties of a dataset, e.g., when the spectrum exhibits a known rate of decay. Our upper and lower bounds for the CSSP/Nyström approximation factor reveal an intriguing phenomenon we call the multiple-descent curve: the approximation factor can exhibit a highly non-monotonic behavior as a function of $k$, with multiple peaks and valleys. These observations suggest a possible connection to the double descent curve exhibited by the generalization error of many machine learning models (see Appendix A for a more in depth discussion of similarities and differences between the two phenomena).

Our analysis technique relies on converting an error bound from random-size DPPs to fixed-size k-DPPs, which results in an additional constant factor of $(1 + 2\epsilon)^2$ in Theorem 1. We put forward a conjecture which would eliminate this factor from Theorem 1 and is of independent interest to the study of elementary symmetric polynomials, a classical topic in combinatorics (Hardy et al., 1952).

**Conjecture 1.** *The following function is* *convex* *with respect to* $k \in [n]$ *for any* $\lambda_1, ..., \lambda_n > 0$:

$$f(k) = (k + 1)\frac{\sum_{S:|S|=k+1} \prod_{i \in S} \lambda_i}{\sum_{S:|S|=k} \prod_{i \in S} \lambda_i}.$$

Deshpande et al. (2006) showed that if $S \sim k\text{-DPP}(\mathbf{A}^\top \mathbf{A})$ and $\lambda_i$ are the eigenvalues of $\mathbf{A}^\top \mathbf{A}$, then $\mathbb{E}[\text{Er}_\mathbf{A}(S)] = f(k)$. If $f(k)$ is convex then Jensen's inequality implies:

$$\mathbb{E}[\text{Er}_\mathbf{A}(S)] \leqslant \mathbb{E}[\text{Er}_\mathbf{A}(S')] \quad \text{for } S' \sim \text{DPP}(\tfrac{1}{\alpha_k} \mathbf{A}^\top \mathbf{A}),$$

where $\alpha_k$ is chosen so that $\mathbb{E}[|S'|] = k$. This would allow us to use the bound from Lemma 1 directly on a k-DPP without relying on the concentration argument of Lemma 2, thereby improving the bounds in Theorems 1 and 2.

### Acknowledgments

We would like to acknowledge DARPA, IARPA (contract W911NF20C0035), NSF, and ONR via its BRC on RandNLA for providing partial support of this work. Our conclusions do not necessarily reflect the position or the policy of our sponsors, and no official endorsement should be inferred.

## Broader Impact

Our study offers a deeper theoretical understanding of the Column Subset Selection Problem. The nature of our work is primarily theoretical, with direct applications to feature selection and kernel approximation, as we noted in Section 1. The primary reason for feature selection as a method for approximating a given matrix, as opposed to a low rank approximation using an SVD, is interpretability, which is crucial in many scientific disciplines. Our analysis shows that in many practical settings, feature selection performs almost as well as SVD at approximating a matrix. As such, our work makes a stronger case for feature selection, wherever applicable, for the sake of interpretability. We also hope our work motivates further research into a fine grained analysis to quantify if machine learning problems are really as hard as worst-case bounds suggest them to be.

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
