[Supplementary Material]

# A   Additional related works

The Column Subset Selection Problem is one of the most classical tasks in matrix approximation (Boutsidis et al., 2008). The original version of the problem compares the projection error of a subset of size $k$ to the best rank $k$ approximation error. The techniques used for finding good subsets have included many randomized methods (Deshpande et al., 2006; Boutsidis et al., 2008; Belhadji et al., 2018; Boutsidis & Woodruff, 2014), as well as deterministic methods (Gu & Eisenstat, 1996). Variants of these algorithms have also been extended to more general losses (Chierichetti et al., 2017; Khanna et al., 2017; Elenberg et al., 2018). Later on, most works have relaxed the problem formulation by allowing the number of selected columns $|S|$ to exceed the rank $k$. These approaches include deterministic sparsification based algorithms (Boutsidis et al., 2011), greedy selection (e.g., Altschuler et al., 2016) and randomized methods (e.g., Drineas et al., 2008; Guruswami & Sinop, 2012; Paul et al., 2015). Note that we study the *original* version of the CSSP (i.e., without the relaxation), where the number of columns $|S|$ must be equal to the rank $k$.

The Nyström method has been given significant attention independently of the CSSP. The guarantees most comparable to our setting are due to Belabbas & Wolfe (2009), who show the approximation factor $k + 1$ for the trace norm error. Many recent works allow the subset size $|S|$ to exceed the target rank $k$, which enables the use of i.i.d. sampling techniques such as leverage scores (Gittens & Mahoney, 2016) and ridge leverage scores (Alaoui & Mahoney, 2015; Musco & Musco, 2017). In addition to the trace norm error, these works consider other types of guarantees, e.g., based on spectral and Frobenius norms, which are not as readily comparable to the CSSP error bounds.

The double descent curve was introduced by Belkin et al. (2019a) to explain the remarkable success of machine learning models which generalize well despite having more parameters than training data. This research has been primarily motivated by the success of deep neural networks, but double descent has also been observed in linear regression (Belkin et al., 2019b; Bartlett et al., 2019; Dereziński et al., 2019b) and other learning models. Double descent is typically presented by plotting the absolute generalization error as a function of the number of parameters used in the learning model, although Poggio et al. (2019) and Liao et al. (2020) showed that the behavior of generalization error is merely an artifact of the phase transitions in the spectral properties of random matrices. Importantly, although the descent curves we obtain are reminiscent of the above works, our setting is different in that it is a *deterministic* combinatorial optimization problem for *relative* error. In particular, Corollary 1 shows that our multiple-descent curve can occur as a purely deterministic property of the optimal CSSP solution. Despite the differences, there are certain similarities between the two settings, namely (a) the notion of stable rank we use matches the one used by Bartlett et al. (2019), (b) the peaks in both the settings are closely aligned – these peaks coincide with the size $k$ crossing the corresponding sharp drops in the respective spectra, (c) the analysis of bias of the minimum norm solution for double descent for linear regression under DPP sampling obtained by Dereziński et al. (2019b) leads to expressions very similar to ours for the CSSP error for DPP sampling.

Determinantal point processes have been shown to provide near-optimal guarantees not only for the CSSP but also other tasks in numerical linear algebra, such as least squares regression (e.g., Avron & Boutsidis, 2013; Dereziński & Warmuth, 2018; Dereziński et al., 2019a). They are also used in recommender systems, stochastic optimization and other tasks in machine learning (for a review, see Dereziński & Mahoney, 2020; Kulesza & Taskar, 2012). Efficient algorithms for sampling from these distributions have been proposed both in the CSSP setting (i.e., given matrix $\mathbf{A}$; see, e.g., Deshpande & Rademacher, 2010; Dereziński, 2019) and in the Nyström setting (i.e., given kernel $\mathbf{K}$; see, e.g., Anari et al., 2016; Dereziński et al., 2019). The term "cardinality constrained DPP" (also known as a "k-DPP" or "volume sampling") was introduced by Kulesza & Taskar (2011) to differentiate from standard DPPs which have random cardinality. Our proofs rely in part on converting DPP bounds to k-DPP bounds via a refinement of the concentration of measure argument used by Dereziński et al. (2020).

Beyond worst-case analysis of algorithms is crucial to understanding the often-noticed gap between practical performance and theoretical guarantees of these algorithms. However, there have been limited number of works in machine learning that undertake finer-grained studies for beyond worst-case analyses. We refer to Roughgarden (2019) for a recent survey of such studies for a few problems in machine learning. Mahoney (2012) takes an alternative view and studies implicit statistical properties of worst case algorithms.

# B   Determinantal point processes

Since our main results rely on randomized subset selection via determinantal point processes (DPPs), we provide a brief overview of the relevant aspects of this class of distributions. First introduced by Macchi (1975), a determinantal point process is a probability distribution over subsets $S \subseteq [n]$, where we use $[n]$ to denote the set $\{1, ..., n\}$. The relative probability of a subset being drawn is governed by a positive semidefinite (p.s.d.) matrix $\mathbf{K} \in \mathbb{R}^{n \times n}$, as stated in the definition below, where we use $\mathbf{K}_{S,S}$ to denote the $|S| \times |S|$ submatrix of $\mathbf{K}$ with rows and columns indexed by $S$.

**Definition 3.** *For an $n \times n$ p.s.d. matrix $\mathbf{K}$, define $S \sim \mathrm{DPP}(\mathbf{K})$ as a distribution over all subsets $S \subseteq [n]$ so that*

$$\Pr(S) = \frac{\det(\mathbf{K}_{S,S})}{\det(\mathbf{I} + \mathbf{K})}.$$

*A restriction to subsets of size $k$ is denoted as $k\text{-}\mathrm{DPP}(\mathbf{K})$.*

DPPs can be used to introduce diversity in the selected set or to model the preference for selecting dissimilar items, where the similarity is stated by the kernel matrix $\mathbf{K}$. DPPs are commonly used in many machine learning applications where these properties are desired, e.g., recommender systems (Warlop et al., 2019), model interpretation (Kim et al., 2016), text and video summarization (Gong et al., 2014), and others (Kulesza & Taskar, 2012). They have also played an important role in randomized numerical linear algebra (Dereziński & Mahoney, 2020).

Given a p.s.d. matrix $\mathbf{K} \in \mathbb{R}^{n \times n}$ with eigenvalues $\lambda_1, ... \lambda_n$, the size of the set $S \sim \mathrm{DPP}(\mathbf{K})$ is distributed as a Poisson binomial random variable, namely, the number of successes in $n$ Bernoulli random trials where the probability of success in the $i$th trial is given by $\frac{\lambda_i}{\lambda_i + 1}$. This leads to a simple expression for the expected subset size:

$$\mathbb{E}[|S|] = \sum_i \frac{\lambda_i}{\lambda_i + 1} = \mathrm{tr}(\mathbf{K}(\mathbf{I} + \mathbf{K})^{-1}). \tag{2}$$

Note that if $S \sim \mathrm{DPP}(\frac{1}{\alpha}\mathbf{K})$, where $\alpha > 0$, then $\Pr(S)$ is proportional to $\alpha^{-|S|} \det(\mathbf{K}_{S,S})$, so rescaling the kernel by a scalar only affects the distribution of the subset sizes, giving us a way to set the expected size to a desired value (larger $\alpha$ means smaller expected size). Nevertheless, it is still often preferrable to restrict the size of $S$ to a fixed $k$, obtaining a $k\text{-}\mathrm{DPP}(\mathbf{K})$ (Kulesza & Taskar, 2011).

Both DPPs and k-DPPs can be sampled efficiently, with some of the first algorithms provided by Hough et al. (2006), Deshpande & Rademacher (2010), Kulesza & Taskar (2011) and others. These approaches rely on an eigendecomposition of the kernel $\mathbf{K}$, at the cost of $O(n^3)$. When $\mathbf{K} = \mathbf{A}^\top \mathbf{A}$, as in the CSSP, and the dimensions satisfy $m \ll n$, then this can be improved to $O(nm^2)$. More recently, algorithms that avoid computing the eigendecomposition have been proposed (Dereziński, 2019; Dereziński et al., 2019; Calandriello et al., 2020; Anari et al., 2016), resulting in running times of $\widetilde{O}(n)$ when given matrix $\mathbf{K}$ and $\widetilde{O}(nm)$ for matrix $\mathbf{A}$, assuming small desired subset size. See Gautier et al. (2019) for an efficient Python implementation of DPP sampling.

The key property of DPPs that enables our analysis is a formula for the expected value of the random matrix that is the orthogonal projection onto the subspace spanned by vectors selected by $\mathrm{DPP}(\mathbf{A}^\top \mathbf{A})$. In the special case when $\mathbf{A}$ is a square full rank matrix, the following result can be derived as a corollary of Theorem 1 by Mutny et al. (2020), and a variant for DPPs over continuous domains can be found as Lemma 8 of Dereziński et al. (2019b). For completeness, we also provide a proof in Appendix C.

**Lemma 5.** *For any $\mathbf{A}$ and $S \subseteq [n]$, let $\mathbf{P}_S$ be the projection onto the $\mathrm{span}\{\mathbf{a}_i : i \in S\}$. If $S \sim \mathrm{DPP}(\mathbf{A}^\top \mathbf{A})$, then*

$$\mathbb{E}[\mathbf{P}_S] = \mathbf{A}(\mathbf{I} + \mathbf{A}^\top \mathbf{A})^{-1} \mathbf{A}^\top.$$

Lemma 5 implies a simple closed form expression for the expected error in the CSSP. Here, we use a rescaling parameter $\alpha > 0$ for controlling the distribution of the subset sizes. Note that it is crucial that we are using a DPP with random subset size, because the corresponding expression for the expected error of the fixed size k-DPP is combinatorial, and therefore much harder to work with.

**Lemma 6.** *For any $\alpha > 0$, if $S \sim \mathrm{DPP}(\frac{1}{\alpha}\mathbf{A}^\top\mathbf{A})$, then*

$$\mathbb{E}\big[\mathrm{Er}_\mathbf{A}(S)\big] = \mathrm{tr}\big(\mathbf{A}\mathbf{A}^\top(\mathbf{I} + \tfrac{1}{\alpha}\mathbf{A}\mathbf{A}^\top)^{-1}\big) = \mathbb{E}[|S|] \cdot \alpha.$$

*Proof.* Using Lemma 5, the expected loss is given by:

$$
\begin{aligned}
\mathbb{E}\big[\mathrm{Er}_\mathbf{A}(S)\big] &= \mathbb{E}\big[\|(\mathbf{I} - \mathbf{P}_S)\mathbf{A}\|_F^2\big] = \mathrm{tr}(\mathbf{A}\mathbf{A}^\top\mathbb{E}[\mathbf{I} - \mathbf{P}_S]) \\
&= \mathrm{tr}\big(\mathbf{A}\mathbf{A}^\top(\mathbf{I} - \tfrac{1}{\alpha}\mathbf{A}(\mathbf{I} + \tfrac{1}{\alpha}\mathbf{A}^\top\mathbf{A})^{-1}\mathbf{A}^\top)\big) \\
&\overset{(*)}{=} \mathrm{tr}\big(\mathbf{A}\mathbf{A}^\top(\mathbf{I} + \tfrac{1}{\alpha}\mathbf{A}\mathbf{A}^\top)^{-1}\big),
\end{aligned}
$$

where $(*)$ follows from the matrix identity $(\mathbf{I} + \mathbf{A}\mathbf{A}^\top)^{-1} = \mathbf{I} - \mathbf{A}(\mathbf{I} + \mathbf{A}^\top\mathbf{A})^{-1}\mathbf{A}^\top$. $\qquad\square$

## C  Proof of Lemma 5

We will use the following standard determinantal summation identity (see Theorem 2.1 in Kulesza & Taskar, 2012) which corresponds to computing the normalization constant $\det(\mathbf{I} + \mathbf{K})$ for a DPP.

**Lemma 7.** *For any $n \times n$ matrix $\mathbf{K}$, we have*

$$\det(\mathbf{I} + \mathbf{K}) = \sum_{S \subseteq [n]} \det(\mathbf{K}_{S,S}).$$

We now proceed with the proof of Lemma 5 (restated below for convenience).

**Lemma' 5.** *For any $\mathbf{A}$ and $S \subseteq [n]$, let $\mathbf{P}_S$ denote the projection onto the $\mathrm{span}\{\mathbf{a}_i : i \in S\}$. If $S \sim \mathrm{DPP}(\mathbf{A}^\top\mathbf{A})$, then*

$$\mathbb{E}[\mathbf{P}_S] = \mathbf{A}(\mathbf{I} + \mathbf{A}^\top\mathbf{A})^{-1}\mathbf{A}^\top.$$

*Proof.* Fix $m$ as the column dimension of $\mathbf{A}$ and let $\mathbf{A}_S$ denote the submatrix of $\mathbf{A}$ consisting of the columns indexed by $S$. We have $\mathbf{P}_S = \mathbf{A}_S(\mathbf{K}_{S,S})^\dagger\mathbf{A}_S$, where $\dagger$ denotes the Moore-Penrose inverse and $\mathbf{K} = \mathbf{A}^\top\mathbf{A}$. Let $\mathbf{v} \in \mathbb{R}^m$ be an arbitrary vector. When $\mathbf{K}_{S,S}$ is invertible, then a standard determinantal identity states that:

$$\det(\mathbf{K}_{S,S})\mathbf{v}^\top\mathbf{P}_S\mathbf{v} = \det(\mathbf{K}_{S,S})\mathbf{v}^\top\mathbf{A}_S\mathbf{K}_{S,S}^{-1}\mathbf{A}_S^\top\mathbf{v} = \det(\mathbf{K}_{S,S} + \mathbf{A}_S^\top\mathbf{v}\mathbf{v}^\top\mathbf{A}_S) - \det(\mathbf{K}_{S,S}).$$

When $\mathbf{K}_{S,S}$ is not invertible then $\det(\mathbf{K}_{S,S}) = \det(\mathbf{K}_{S,S} + \mathbf{A}_S^\top\mathbf{v}\mathbf{v}^\top\mathbf{A}_S) = 0$, because the rank of $\mathbf{K}_{S,S} + \mathbf{A}_S^\top\mathbf{v}\mathbf{v}^\top\mathbf{A}_S = \mathbf{A}_S^\top(\mathbf{I} + \mathbf{v}\mathbf{v}^\top)\mathbf{A}_S$ cannot be higher than the rank of $\mathbf{K}_{S,S} = \mathbf{A}_S^\top\mathbf{A}_S$. Thus,

$$
\begin{aligned}
\det(\mathbf{I} + \mathbf{K})\mathbf{v}^\top\mathbb{E}[\mathbf{P}_S]\mathbf{v} &= \sum_{S \subseteq [n]:\, \det(\mathbf{K}_{S,S})>0} \det(\mathbf{K}_{S,S})\mathbf{v}^\top\mathbf{A}_S\mathbf{K}_{S,S}^{-1}\mathbf{A}_S^\top\mathbf{v} \\
&= \sum_{S \subseteq [n]} \det(\mathbf{K}_{S,S} + \mathbf{A}_S^\top\mathbf{v}\mathbf{v}^\top\mathbf{A}_S) - \det(\mathbf{K}_{S,S}) \\
&= \sum_{S \subseteq [n]} \det\big([\mathbf{K} + \mathbf{A}^\top\mathbf{v}\mathbf{v}^\top\mathbf{A}]_{S,S}\big) - \sum_{S \subseteq [n]} \det(\mathbf{K}_{S,S}) \\
&\overset{(*)}{=} \det(\mathbf{I} + \mathbf{K} + \mathbf{A}^\top\mathbf{v}\mathbf{v}^\top\mathbf{A}) - \det(\mathbf{I} + \mathbf{K}) \\
&= \det(\mathbf{I} + \mathbf{K})\mathbf{v}^\top\mathbf{A}(\mathbf{I} + \mathbf{K})^{-1}\mathbf{A}^\top\mathbf{v},
\end{aligned}
$$

where $(*)$ involves two applications of Lemma 7. Since the above calculation holds for arbitrary vector $\mathbf{v}$, the claim follows. $\qquad\square$

## D  Proofs omitted from Section 2

**Lemma' 1.** *For any $\mathbf{A}$, $0 \leqslant \epsilon < 1$ and $s < k < t_s$, where $t_s = s + \mathrm{sr}_s(\mathbf{A})$, suppose that $S \sim \mathrm{DPP}(\frac{1}{\alpha}\mathbf{A}^\top\mathbf{A})$ for $\alpha = \frac{\gamma_s(k)\mathrm{OPT}_k}{(1-\epsilon)(k-s)}$ and $\gamma_s(k) = \sqrt{1 + \frac{2(k-s)}{t_s-k}}$. Then:*

$$\frac{\mathbb{E}\big[\mathrm{Er}_\mathbf{A}(S)\big]}{\mathrm{OPT}_k} \leqslant \frac{\Phi_s(k)}{1-\epsilon} \quad and \quad \mathbb{E}[|S|] \leqslant k - \epsilon\,\frac{k-s}{\gamma_s(k)},$$

*where $\Phi_s(k) = \big(1 + \frac{s}{k-s}\big)\gamma_s(k)$.*

*Proof.* Let $\lambda_1 \geqslant \lambda_2 \geqslant ...$ be the eigenvalues of $\mathbf{A}^\top \mathbf{A}$. Note that scaling the matrix $\mathbf{A}$ by any constant $c$ and scaling $\alpha$ by $c^2$ preserves the distribution of $S$ as well as the approximation ratio, so without loss of generality, assume that $\lambda_{s+1} = 1$. Furthermore, using the shorthands $l = k - s$ and $r = \mathrm{sr}_s(\mathbf{A})$, we have $t_s - k = r - l$ and so $\gamma_s(k) = \sqrt{\frac{r+l}{r-l}}$. We now lower bound the optimum as follows:

$$\mathrm{OPT}_k = \sum_{j>k} \lambda_j = \mathrm{sr}_s(\mathbf{A}) - \sum_{j=s+1}^{k} \lambda_j \geqslant r - l.$$

We will next define an alternate sequence of eigenvalues which is in some sense "worst-case", by shifting the spectral mass away from the tail. Let $\lambda'_{s+1} = ... = \lambda'_k = 1$, and for $i > k$ set $\lambda'_i = \beta\lambda_i$, where $\beta = \frac{r-l}{\mathrm{OPT}_k} \leqslant 1$. Additionally, define:

$$\alpha' = \beta\alpha = \frac{\gamma_s(k)(r-l)}{(1-\epsilon)l} = \frac{\sqrt{r^2 - l^2}}{(1-\epsilon)l},$$

$$\alpha'' = (1-\epsilon)\frac{\sqrt{r+l} + \sqrt{r-l}}{2\sqrt{r+l}}\alpha' = \frac{(\sqrt{r+l} + \sqrt{r-l})\sqrt{r-l}}{r+l-(r-l)} = \frac{\sqrt{r-l}}{\sqrt{r+l} - \sqrt{r-l}}. \quad (3)$$

and note that $\alpha'' \leqslant \alpha' \leqslant \alpha$. Moreover, for $s + 1 \leqslant i \leqslant k$, we let $\alpha'_i = \alpha''$, while for $i > k$ we set $\alpha'_i = \alpha'$. We proceed to bound the expected subset size $\mathbb{E}[|S|]$ by converting all the eigenvalues from $\lambda_i$ to $\lambda'_i$ and $\alpha$ to $\alpha'_i$, which will allow us to easily bound the entire expression:

$$\mathbb{E}[|S|] = \sum_i \frac{\lambda_i}{\lambda_i + \alpha} \leqslant s + \sum_{i=s+1}^{k} \frac{\lambda_i}{\lambda_i + \alpha'_i} + \sum_{i>k} \frac{\beta\lambda_i}{\beta\lambda_i + \beta\alpha} \leqslant s + \sum_{i=s+1}^{k} \frac{\lambda'_i}{\lambda'_i + \alpha''} + \sum_{i>k} \frac{\lambda'_i}{\lambda'_i + \alpha'}. \quad (4)$$

We bound each of the two sums separately starting with the first one:

$$\sum_{i=s+1}^{k} \frac{\lambda'_i}{\lambda'_i + \alpha''} = \frac{l}{1+\alpha''} = l - \frac{l}{1+\alpha''} = l - \frac{l}{1 + \frac{1}{\alpha''}} = l - \frac{l}{1 + \frac{\sqrt{r+l} - \sqrt{r-l}}{\sqrt{r-l}}} = l - \frac{l\sqrt{r-l}}{\sqrt{r+l}}. \quad (5)$$

To bound the second sum, we use the fact that $\sum_{i>k} \lambda'_i = \beta\,\mathrm{OPT}_k = r - l$, and obtain:

$$\sum_{i>k} \frac{\lambda'_i}{\lambda'_i + \alpha'} \leqslant \frac{1}{\alpha'} \sum_{i>k} \lambda'_i = \frac{r-l}{\alpha'} = (1-\epsilon)\frac{l\sqrt{r-l}}{\sqrt{r+l}}. \quad (6)$$

Combining the two sums, we conclude that $\mathbb{E}[|S|] \leqslant s + l - \epsilon\,l\sqrt{\frac{r-l}{r+l}} = k - \frac{\epsilon l}{\gamma_s(k)}$. Finally, Lemma 6 yields:

$$\frac{\mathbb{E}[\mathrm{Er}_\mathbf{A}(S)]}{\mathrm{OPT}_k} = \frac{\mathbb{E}[|S|] \cdot \alpha}{\mathrm{OPT}_k} \leqslant \frac{k}{k-s}\frac{\gamma_s(k)}{1-\epsilon} = \frac{\Phi_s(k)}{1-\epsilon},$$

which concludes the proof. □

**Lemma' 2.** *Let $S$ be sampled as in Lemma 1 with $\epsilon \leqslant \frac{1}{2}$. If $s + \frac{7}{\epsilon^4}\ln^2\frac{1}{\epsilon} \leqslant k \leqslant t_s - 1$, then $\Pr(|S| > k) \leqslant \epsilon$.*

*Proof.* Let $p_i = \frac{\lambda'_i}{\lambda'_i + \alpha'_i}$ be the Bernoulli probabilities for $b_i \sim \mathrm{Bernoulli}(p_i)$ and $X = \sum_{i>s} b_i$, where $\lambda'_i$ and $\alpha'_i$ are as defined in the proof of Lemma 1. Note that $|S|$ is distributed as a Poisson binomial random variable such that the success probability associated with the $i$th eigenvalue is upper-bounded by $p_i$ for each $i > s$. It follows that $\Pr(|S| > k) \leqslant \Pr(X > l)$, where $l = k - s$. Moreover, letting $r = \mathrm{sr}_s(\mathbf{A})$, in the proof of Lemma 1 we showed that:

$$k - \mathbb{E}[X] \geqslant \epsilon\frac{l\sqrt{r-l}}{\sqrt{r+l}},$$

and furthermore, using the derivations in (5) and (6) together with the formula $\mathrm{Var}[b_i] = p_i(1-p_i)$, we obtain that:

$$\mathrm{Var}[X] \leqslant \sum_{i=s+1}^{k} (1-p_i) + \sum_{i>k} p_i \leqslant \frac{l\sqrt{r-l}}{\sqrt{r+l}} + (1-\epsilon)\frac{l\sqrt{r-l}}{\sqrt{r+l}} = (2-\epsilon)\frac{l\sqrt{r-l}}{\sqrt{r+l}}.$$

Using Theorem 2.6 from Chung & Lu (2006) with $\lambda = \epsilon \frac{l\sqrt{r-l}}{\sqrt{r+l}}$, we have:

$$\Pr(|S| > k) \leqslant \Pr(X > l) \leqslant \Pr(X > \mathbb{E}[X] + \lambda) \leqslant \exp\Big(-\frac{\lambda^2}{2(\mathrm{Var}[X] + \lambda/3)}\Big)$$

$$\leqslant \exp\Big(-\frac{\lambda^2}{2(\frac{2-\epsilon}{\epsilon}\lambda + \lambda/3)}\Big) \leqslant \exp(-\epsilon\lambda/4) = \exp\Big(-\frac{\epsilon^2 l\sqrt{r-l}}{4\sqrt{r+l}}\Big).$$

Note that since $7 \leqslant l \leqslant r - 1$, we have $l\frac{\sqrt{r-l}}{\sqrt{r+l}} \geqslant \frac{l}{\sqrt{2l+1}} \geqslant \frac{7}{16}\sqrt{l}$, so by simple algebra it follows that for $l \geqslant \frac{7}{\epsilon^4}\ln^2\frac{1}{\epsilon}$, we have $l\frac{\sqrt{r-l}}{\sqrt{r+l}} \geqslant \frac{4}{\epsilon^2}\ln\frac{1}{\epsilon}$ and therefore $\Pr(|S| > k) \leqslant \epsilon$. $\qquad\square$

**Lemma' 3.** *For any* $\mathbf{A} \in \mathbb{R}^{m \times n}$, $k \in [n]$ *and* $\alpha > 0$, *if* $S \sim \mathrm{DPP}(\frac{1}{\alpha}\mathbf{A}^\top\mathbf{A})$ *and* $S' \sim k\text{-DPP}(\mathbf{A}^\top\mathbf{A})$, *then*

$$\mathbb{E}\big[\mathrm{Er}_\mathbf{A}(S')\big] \leqslant \mathbb{E}\big[\mathrm{Er}_\mathbf{A}(S) \mid |S| \leqslant k\big].$$

*Proof.* Let $\lambda_1 \geqslant \lambda_2 \geqslant \dots$ denote the eigenvalues of $\mathbf{A}^\top\mathbf{A}$ and let $e_k$ be the $k$th elementary symmetric polynomial of $\mathbf{A}$:

$$e_k = \sum_{T:|T|=k} \det(\mathbf{A}_T^\top\mathbf{A}_T) = \sum_{T:|T|=k}\prod_{i\in T}\lambda_i.$$

Also let $\bar{e}_k = e_k/\binom{n}{k}$ denote the $k$th elementary symmetric mean. Newton's inequalities imply that:

$$1 \geqslant \frac{\bar{e}_{k-1}\bar{e}_{k+1}}{\bar{e}_k^2} = \frac{e_{k-1}e_{k+1}}{e_k^2}\frac{\binom{n}{k}}{\binom{n}{k-1}}\frac{\binom{n}{k}}{\binom{n}{k+1}} = \frac{e_{k-1}e_{k+1}}{e_k^2}\frac{n+1-k}{k}\frac{k+1}{n-k}.$$

The results of Deshpande et al. (2006) and Guruswami & Sinop (2012) establish that $\mathbb{E}[\mathrm{Er}_\mathbf{A}(S) \mid |S| = k] = (k+1)\frac{e_{k+1}}{e_k}$, so it follows that:

$$\frac{\mathbb{E}[\mathrm{Er}_\mathbf{A}(S) \mid |S| = k]}{\mathbb{E}[\mathrm{Er}_\mathbf{A}(S) \mid |S| = k-1]} = \frac{k+1}{k}\frac{e_{k+1}e_{k-1}}{e_k^2} \leqslant \frac{n-k}{n+1-k} \leqslant 1. \qquad (7)$$

Finally, note that $\mathbb{E}[\mathrm{Er}_\mathbf{A}(S) \mid |S| \leqslant k]$ is a weighted average of components $\mathbb{E}[\mathrm{Er}_\mathbf{A}(S) \mid |S| = s]$ for $s \in [k]$, and (7) implies that the smallest of those components is associated with $s = k$. Since the weighted average is lower bounded by the smallest component, this completes the proof. $\qquad\square$

# E  Proof of Theorem 2

Before showing Theorem 2, we give an additional lemma which covers the corner case of the theorem when $k$ is close to $n$.

**Lemma 8.** *Given* $\mathbf{A} \in \mathbb{R}^{m \times n}$ *and* $s < k < n$, *let* $\lambda_1 \geqslant \dots \geqslant \lambda_n > 0$ *be the eigenvalues of* $\mathbf{A}^\top\mathbf{A}$. *If* $S \sim k\text{-DPP}(\mathbf{A}^\top\mathbf{A})$ *and we let* $b = \min\{k-s, n-k\}$, *then for any* $0 < \epsilon \leqslant \frac{1}{2}$ *we have*

$$\frac{\mathbb{E}[\mathrm{Er}_\mathbf{A}(S)]}{\mathrm{OPT}_k} \leqslant \big(1 - e^{-\frac{\epsilon^2 b}{10}}\big)^{-1}(1-\epsilon)^{-1}\Psi_s(k),$$

*where* $\Psi_s(k) = \frac{\lambda_{s+1}}{\lambda_n}\big(1 + \frac{s}{k-s}\big)$.

*Proof.* Let $\alpha = \frac{\lambda_{s+1}}{(1-\epsilon)\lambda_n} \frac{\text{OPT}_k}{k-s}$. Note that $\text{OPT}_k = \sum_{i>k} \lambda_i \geqslant (n-k)\lambda_n$. Define $b_i \sim$ Bernoulli$(\frac{\lambda_i}{\lambda_i+\alpha})$ and let $X = \sum_{i>s} b_i$. We have:

$$\mathbb{E}[X] = \sum_{i>s} \frac{\lambda_i}{\lambda_i + \alpha}$$

$$\leqslant \frac{(n-s)\lambda_{s+1}}{\lambda_{s+1} + \frac{\lambda_{s+1}}{\lambda_n}\frac{(n-k)\lambda_n}{(1-\epsilon)(k-s)}}$$

$$= \frac{1}{\frac{1}{n-s} + \frac{1}{(1-\epsilon)(k-s)}\frac{n-k}{n-s}}$$

$$= \frac{1}{\frac{1}{n-s} + \frac{1}{(1-\epsilon)(k-s)}\left(1 - \frac{k-s}{n-s}\right)}$$

$$= \frac{1}{\frac{1}{(1-\epsilon)(k-s)} - \frac{\epsilon}{1-\epsilon}\frac{1}{n-s}}$$

$$= \frac{1-\epsilon}{\frac{1}{k-s} - \frac{\epsilon}{n-s}}.$$

Let $S' \sim \text{DPP}(\frac{1}{\alpha}\mathbf{A}^\top\mathbf{A})$. It follows that

$$k - \mathbb{E}[|S'|] \geqslant k - (s + \mathbb{E}[X])$$

$$\geqslant (k-s) - \frac{1-\epsilon}{\frac{1}{k-s} - \frac{\epsilon}{n-s}}$$

$$= (k-s)\left(1 - \frac{1-\epsilon}{1 - \epsilon\frac{k-s}{n-s}}\right)$$

$$= (k-s)\frac{\epsilon - \epsilon\frac{k-s}{n-s}}{1 - \epsilon\frac{k-s}{n-s}}$$

$$\geqslant \epsilon(k-s)\left(1 - \frac{k-s}{n-s}\right)$$

$$= \epsilon \cdot \frac{(k-s)(n-k)}{n-s}$$

$$\geqslant \frac{\epsilon}{2} \cdot \min\{k-s, n-k\}.$$

From this, it follows that:

$$\frac{\mathbb{E}[\text{Er}_\mathbf{A}(S')]}{\text{OPT}_k} = \frac{\mathbb{E}[|S|] \cdot \alpha}{\text{OPT}_k} \leqslant (1-\epsilon)^{-1}\frac{k}{k-s}\frac{\lambda_{s+1}}{\lambda_n} = (1-\epsilon)^{-1}\left(1 + \frac{s}{k-s}\right)\frac{\lambda_{s+1}}{\lambda_n}.$$

We now give an upper bound on $\Pr(|S'| > k)$ by considering two cases.

**Case 1**: $k-s \leqslant n-k$. Then, using $\lambda = \epsilon(k-s)/2$, we have $(k-s) - \mathbb{E}[X] \geqslant \lambda$, so using Theorem 2.4 from Chung & Lu (2006), we get:

$$\Pr(|S'| > k) \leqslant \Pr(X > k-s) \leqslant \Pr(X > \mathbb{E}[X] + \lambda) \leqslant e^{-\frac{\lambda^2}{2(k-s)}} = e^{-\epsilon^2(k-s)/8}.$$

**Case 2**: $k - s > n - k$. Then, using Theorem 2.4 from Chung & Lu (2006) with $\lambda = k - \mathbb{E}[|S'|] = \frac{\epsilon(n-k)}{2} + \Delta$, where $\Delta > 0$, we get:

$$\Pr(|S'| > k) = \Pr(n - |S'| < n - k)$$

$$\leqslant \exp\left(-\frac{\lambda^2}{2\,\mathbb{E}[n-|S'|]}\right)$$

$$= \exp\left(-\frac{\lambda}{2}\frac{\frac{\epsilon}{2}(n-k) + \Delta}{n - k + \frac{\epsilon}{2}(n-k) + \Delta}\right)$$

$$\leqslant \exp\left(-\frac{\lambda}{2}\frac{\frac{\epsilon}{2}(n-k)}{n - k + \frac{\epsilon}{2}(n-k)}\right)$$

$$= \exp\left(-\frac{\epsilon^2(n-k)}{8(1 + \epsilon/2)}\right)$$

$$\overset{(*)}{\leqslant} \exp\left(-\frac{\epsilon^2(n-k)}{10}\right),$$

where in $(*)$ we used the fact that $\epsilon \in (0, \frac{1}{2})$. Now, the result follows easily by invoking Lemma 3:

$$\mathbb{E}\big[\mathrm{Er}_{\mathbf{A}}(S)\big] \leqslant \mathbb{E}\big[\mathrm{Er}_{\mathbf{A}}(S') \mid |S'| \leqslant k\big] \leqslant \frac{\mathbb{E}\big[\mathrm{Er}_{\mathbf{A}}(S')\big]}{\Pr(|S'| \leqslant k)}$$

$$\leqslant \big(1 - \mathrm{e}^{-\frac{\epsilon^2 b}{10}}\big)^{-1}(1-\epsilon)^{-1}\frac{\lambda_{s+1}}{\lambda_n}\Big(1 + \frac{s}{k-s}\Big) \cdot \mathrm{OPT}_k,$$

which completes the proof. $\square$

Note that since $b \geqslant 1$, setting $\epsilon = \frac{1}{2}$ in Lemma 8 yields the following simpler (but usually much weaker) bound:

$$\frac{\mathbb{E}[\mathrm{Er}_{\mathbf{A}}(S)]}{\mathrm{OPT}_k} \leqslant 2\big(1 - \mathrm{e}^{-\frac{1}{40}}\big)^{-1}\Psi_s(k) \leqslant 82\,\Psi_s(k).$$

**Theorem' 2.** *Let $\lambda_1 \geqslant \lambda_2 \geqslant \dots$ be the eigenvalues of $\mathbf{A}^\top \mathbf{A}$. There is an absolute constant $c$ such that for any $0 < c_1 \leqslant c_2$, with $\gamma = c_2/c_1$, if:*

1. *(**polynomial spectral decay**) $c_1 i^{-p} \leqslant \lambda_i \leqslant c_2 i^{-p}$ $\forall_i$, with $p > 1$, then $S \sim k\text{-DPP}(\mathbf{A}^\top \mathbf{A})$ satisfies*

$$\frac{\mathbb{E}[\mathrm{Er}_{\mathbf{A}}(S)]}{\mathrm{OPT}_k} \leqslant c\gamma p.$$

2. *(**exponential spectral decay**) $c_1(1-\delta)^i \leqslant \lambda_i \leqslant c_2(1-\delta)^i$ $\forall_i$, with $\delta \in (0,1)$, then $S \sim k\text{-DPP}(\mathbf{A}^\top \mathbf{A})$ satisfies*

$$\frac{\mathbb{E}[\mathrm{Er}_{\mathbf{A}}(S)]}{\mathrm{OPT}_k} \leqslant c\gamma(1 + \delta k).$$

*Proof.* **(1) Polynomial decay.** We provide the proof by splitting it into two cases.

**Case 1(a)**: $\left(\frac{k+1}{n}\right)^{p-1} \leqslant \frac{1}{2}$

We can use upper and lower integrals to bound the sum $\sum_{i \geqslant s} \frac{1}{i^p}$ as:

$$\int_{x \geqslant (s+1)} \frac{1}{i^p}dx \leqslant \sum_{i \geqslant s} \frac{1}{i^p} \leqslant \int_{x \geqslant s} \frac{1}{i^p}dx \implies \sum_{i=s+1}^{n} \frac{1}{i^p} \geqslant \frac{(s+2)^{1-p}}{p-1} - \frac{(n+1)^{1-p}}{p-1}.$$

We lower bound the stable rank for $s \leqslant k$ using the upper/lower bounds on the eigenvalues and the condition for Case 1(a):

$$\mathrm{sr}_s(\mathbf{A}) = \frac{\sum_{i=s+1}^{n} \lambda_i}{\lambda_{s+1}}$$

$$\geqslant \frac{c_1}{c_2} \left( \frac{(s+2)^{1-p}}{(p-1)(s+1)^{-p}} - \frac{(n+1)^{1-p}}{(p-1)(s+1)^{-p}} \right)$$

$$= \frac{1}{\gamma} \left( \frac{s+2}{p-1} \left(1 - \frac{1}{s+2}\right)^p - \frac{s+1}{p-1} \left(\frac{s+1}{n+1}\right)^{p-1} \right)$$

$$\geqslant \frac{1}{\gamma} \left( \frac{s+2}{p-1} - 1 - \frac{s+1}{p-1} \cdot \frac{1}{2} \right) = \frac{1}{2\gamma} \frac{s+1}{p-1} - \frac{1}{\gamma}.$$

Further using $u = k - s$, we can call upon Theorem 1 to get,

$$\Phi_s(k) \leqslant \frac{k}{u}\sqrt{1 + \frac{2u}{\mathrm{sr}_s - u}} \leqslant \frac{k}{u} + \frac{k}{\frac{1}{2\gamma}\frac{s+1}{p-1} - \gamma^{-1} - u} = \frac{k}{u} + \frac{(2p-2)k}{\gamma^{-1}(s+1-2p+2) - (2p-2)u}$$

$$\leqslant \frac{k}{u} + \frac{(2p-2+\gamma^{-1})k}{\gamma^{-1}(k+3-2p) - (2p-2+\gamma^{-1})u}$$

Optimizing over $u$, we see that the minimum is reached for $u = \hat{u} = \frac{k+3-2p}{2\gamma(2p-2+\gamma^{-1})}$ which achieves the value $\frac{4(\gamma(2p-2)+1)k}{k+3-2p}$ which is upper bounded by $\frac{12\gamma pk}{(k-2p)}$.

We assume $k \geqslant \hat{u} > 60p > 60$. If not, Deshpande et al. (2006) ensure an upper bound of $(k+1) \leqslant 60p + 1 < 61p$. With $p < k/60$, we get:

$$\frac{12\gamma pk}{k-2p} \leqslant \frac{12\gamma pk}{k - k/30} = \frac{12\gamma p}{1 - 1/30} \leqslant \frac{360}{29}\gamma p.$$

Since we assumed that $\hat{u} > 60$, then $k - s > \frac{7}{\epsilon^4} \ln^2 \frac{1}{\epsilon}$ for $\epsilon = 0.5$ which means $(1+2\epsilon)^2 \leqslant 4$, which makes the approximation ratio upper bounded by $\frac{1440}{29}\gamma p$. The overall bound thus becomes $61\gamma p$.

**Case 1(b):** $\left(\frac{k+1}{n}\right)^{p-1} > \frac{1}{2}$

From Lemma 8, we know that the approximation ratio is upper bounded by constant factor times $\Psi_s(k) = \frac{\lambda_{s+1}}{\lambda_n} \frac{k}{k-s}$. Consider,

$$\Psi_s(k) = \frac{\lambda_{s+1}}{\lambda_n} \frac{k}{k-s} \leqslant \gamma \frac{n^p}{(s+1)^p} \frac{k}{k-s} = \gamma \left(\frac{n}{k+1}\right)^{p-1} \frac{k+1}{n} \frac{(k+1)^p}{(s+1)^p} \frac{k}{k-s} \leqslant 2\gamma \left(\frac{k+1}{s+1}\right)^p \frac{k}{k-s},$$

which holds true for all $s \leqslant k$, and is optimized for $s = \hat{s} = \frac{pk-1}{p+1}$. We get that the approximation ratio is bounded as:

$$\Psi_s(k) \leqslant \gamma \frac{k(p+1)}{k+1} \left(\frac{p+1}{p}\right)^p \leqslant e\gamma(p+1) \leqslant 2e\gamma p.$$

Combining in the factor based on $\epsilon$ in Lemma 8, we get an upper bound of $164e\gamma p$ that is larger than the bound obtained in the case 1(a) above and hence covers all the subcases.

### (2) Exponential decay.

We first lower bound the stable rank of $\mathbf{A}$ of order $s$:

$$\mathrm{sr}_s(\mathbf{A}) = \sum_{j>s} \lambda_j / \lambda_{s+1} \geqslant \frac{c_1(1 - (1-\delta)^{n-s})/\delta}{c_2} = \frac{1 - (1-\delta)^{n-s}}{\gamma\delta}.$$

We present the proof by considering two subcases separately : when $k \leqslant n - \frac{\ln 2}{\delta}$ and $k > n - \frac{\ln 2}{\delta}$.

**Case 2(a):** $k \leqslant n - \frac{\ln 2}{\delta}$. From the assumption, letting $s \leqslant k$ we have

$$s \leqslant n - \frac{\ln 2}{\delta}$$

$$\implies s \leqslant n - \frac{\ln 2}{\ln \frac{1}{1-\delta}}$$

$$\implies (n - s) \ln \frac{1}{1-\delta} \geqslant \ln 2$$

$$\iff 1 - (1 - \delta)^{n-s} \geqslant \frac{1}{2}$$

$$\implies \mathrm{sr}_s(\mathbf{K}) \geqslant \frac{1}{2\gamma\delta},$$

where the second inequality follows because $\frac{x}{1+x} \leqslant \ln(1 + x)$ with $x = \delta/(1 - \delta)$.

We will use $u = k - s$. From Theorem 1, using $\mathrm{sr}_s \geqslant \frac{1}{2\gamma\delta}$ we have the following upper bound:

$$\Phi_s(k) \leqslant \frac{k}{u} \left( 1 + \frac{2\gamma\delta u}{1 - 2\gamma\delta u} \right) = \frac{k}{u} \cdot \frac{1}{1 - 2\gamma\delta u}.$$

RHS is minimized for $\hat{u} = \frac{1}{4\gamma\delta}$. We let $\epsilon = 0.5$ and assume that $\hat{u} \geqslant 60$ which is bigger than $\frac{7}{\epsilon^4} \ln^2 \frac{1}{\epsilon}$. If not, then $\delta \geqslant \frac{4}{60\gamma} > \frac{1}{\gamma}$ and the worst-case bound of Deshpande et al. (2006) ensures that the approximation factor is no more than $k + 1 \leqslant \gamma(1 + \frac{1}{\gamma}k) \leqslant \gamma(1 + \delta k)$. By a similar argument we can assume that $k \geqslant 60$.

If $k \leqslant \hat{u}$, in this case we can set $s = 0$, i.e., $u = k$, obtaining $\Phi_s(k) \leqslant \frac{1}{1-2\gamma\delta k} \leqslant 2$. And so the approximation ratio is bounded by $(1 + 2\epsilon)^2 \cdot 2 \leqslant 8$. On the other hand, if $k > \hat{u}$, we can set $u = \hat{u}$, which implies $\Phi_s(k) \leqslant 8\gamma\delta k$, and so the approximation ratio is bounded by $32\gamma\delta k$. The overall bound is thus $61\gamma(1 + \delta k)$ covering all possible subcases.

**Case 2(b):** $k > n - \frac{\ln 2}{\delta}$. We make use of Lemma 8 for the case when $k$ is close to $n$. The approximation guarantee uses:

$$\Psi_s(k) = \frac{\lambda_{s+1}}{\lambda_n} \frac{k}{k - s},$$

where $s < k$. For our bound, we choose $s = \lfloor k - \frac{\ln 2}{\delta} \rfloor$. This implies that $n - s < \frac{2 \ln 2}{\delta} + 1 = \frac{\delta + \ln 4}{\delta}$. It follows that

$$\frac{\lambda_{s+1}}{\lambda_n} \leqslant \frac{\gamma}{(1-\delta)^{n-s}} \leqslant \frac{\gamma}{(1-\delta)^{(\delta+\ln 4)/\delta}} = \gamma \left[ (1 - \delta)^{-\frac{1}{\delta}} \right]^{\delta + \ln 4} \leqslant \gamma e^{\frac{\delta + \ln 4}{1 - \delta}}.$$

If $\delta \geqslant \frac{1}{20}$, then the worst-case result of Deshpande et al. (2006) suffices to show that the approximation ratio is bounded by $k + 1 \leqslant 20(1 + \delta k)$, so assume that $\delta < \frac{1}{20}$. Then we have $e^{\frac{\delta + \ln 4}{1 - \delta}} < 5$. Combining this with the fact that $\frac{k}{k-s} \leqslant \frac{\delta k}{\ln 2}$, we obtain:

$$\Phi_s(k) \leqslant \frac{5\gamma\delta k}{\ln 2}.$$

Combining with factor based on $\epsilon$ in Lemma 8, we get $82 \cdot \frac{5\gamma\delta k}{\ln 2}$. Thus, the bound of $\frac{82 \cdot 5}{\ln 2} \gamma(1 + \delta k)$ holds in all cases, completing the proof. $\qquad \square$

# F    Proof of Lemma 4

**Lemma' 4.** *Fix $\delta \in (0, 1)$ and consider unit vectors $\mathbf{a}_{i,j} \in \mathbb{R}^m$ in general position, where $i \in [t]$, $j \in [l_i]$, such that $\sum_j \mathbf{a}_{i,j} = 0$ for each $i$, and for any $i, j, i', j'$, if $i \neq i'$ then $\mathbf{a}_{i,j}$ is orthogonal to*

$\mathbf{a}_{i',j'}$. Also, let unit vectors $\{\mathbf{v}_i\}_{i\in[t]}$ be orthogonal to each other and to all $\mathbf{a}_{i,j}$. There are positive scalars $\alpha_i, \beta_i$ for $i \in [t]$ such that matrix $\mathbf{A}$ with columns $\alpha_i \mathbf{a}_{i,j} + \beta_i \mathbf{v}_i$ over all $i$ and $j$ satisfies:

$$\min_{|S|=k_i} \frac{\mathrm{Er}_\mathbf{A}(S)}{\mathrm{OPT}_{k_i}} \geqslant (1-\delta)l_i, \quad for\ k_i = l_1 + ... + l_i - 1.$$

*Proof.* Say $\widehat{\mathbf{A}}_i$ is the matrix obtained by stacking all the $\mathbf{a}_{i,j}$ and let $\lambda_{i,1} \geqslant \lambda_{i,2} \geqslant ... \geqslant \lambda_{i,l_i-1}$ denote the non-zero eigenvalues of $\widehat{\mathbf{A}}_i^\top \widehat{\mathbf{A}}_i$. We write $\tilde{\mathbf{a}}_{i,j} = \alpha_i \mathbf{a}_{i,j} + \beta_i \mathbf{v}_i$ and note that for each $i$, $\mathbf{1}_{l_i}$ is an eigenvector of $\widehat{\mathbf{A}}_i^\top \widehat{\mathbf{A}}_i$ with eigenvalue 0. Further, $\mathbf{A}^\top \mathbf{A}$ is a block-diagonal matrix with blocks $\mathbf{B}_i = \alpha_i^2 \widehat{\mathbf{A}}_i^\top \widehat{\mathbf{A}}_i + \beta_i^2 \mathbf{1}_{l_i} \mathbf{1}_{l_i}^\top$:

$$\mathbf{A}^\top \mathbf{A} = \begin{bmatrix} \mathbf{B}_1 & \mathbf{0} & \mathbf{0} \\ \mathbf{0} & \ddots & \mathbf{0} \\ \mathbf{0} & \mathbf{0} & \mathbf{B}_t \end{bmatrix}$$

Therefore, the eigenvalues of $\mathbf{A}^\top \mathbf{A}$ are $\alpha_1^2 \lambda_{1,1}, ..., \alpha_1^2 \lambda_{1,l_1-1}, \beta_1^2 l_1, ..., \alpha_t^2 \lambda_{t,1}, ..., \alpha_t^2 \lambda_{t,l_t-1}, \beta_t^2 l_t$, and so we can always choose the parameters so that $\alpha_i \gg \beta_i \gg \alpha_{i+1}$ for each $i$, ensuring that these eigenvalues are in decreasing order. Let us fix an arbitrary $c \in [t]$. From the above, it follows that for $k_c = \left(\sum_{i\leqslant c} l_i\right) - 1$ we have:

$$\mathrm{OPT}_{k_c} = l_c \beta_c^2 + \sum_{i>c} \mathrm{tr}(\mathbf{B}_i) = l_c \beta_c^2 + \phi_c,$$

where we use $\phi_c = \sum_{i>c} \mathrm{tr}(\mathbf{B}_i)$ as a shorthand. Since the centroid of $\{\tilde{\mathbf{a}}_{c,1}, \ldots, \tilde{\mathbf{a}}_{c,l_c}\}$ is $\beta \mathbf{v}_c$, we can write $\tilde{\mathbf{a}}_{c,l_c} = l_c \beta \mathbf{v}_c - \sum_{j<l_c} \tilde{\mathbf{a}}_{c,j}$. For selecting the set $S \subset [n]$ of size $k_c$, since $\alpha_i \gg \alpha_{i+1}$, we can assume without loss of generality that $S$ does not select any vectors $\tilde{\mathbf{a}}_{i,j}$ such that $i > c$ and does not drop any such that $i < c$, and so for some $j' \in [l_c]$ we let $S_{j'}$ be the index set such that $\mathbf{P}_{S_{j'}}$ is the projection onto the span of $\left(\bigcup_{i<c} \bigcup_j \{\tilde{\mathbf{a}}_{i,j}\}\right) \cup \{\tilde{\mathbf{a}}_{c,1}, \ldots, \tilde{\mathbf{a}}_{c,l_c}\} \backslash \{\tilde{\mathbf{a}}_{c,j'}\}$. We now lower bound the squared projection error of that set:

$$\mathrm{Er}_\mathbf{A}(S_{j'}) = \|\tilde{\mathbf{a}}_{c,j'} - \mathbf{P}_{S_{j'}} \tilde{\mathbf{a}}_{c,j'}\|^2 + \sum_{i>c} \sum_{j=1}^{l_i} \|\tilde{\mathbf{a}}_{i,j} - \mathbf{P}_{S_{j'}} \tilde{\mathbf{a}}_{i,j}\|^2$$

$$= \left\| l_c \beta \mathbf{v}_c - \sum_{j<l_c} \tilde{\mathbf{a}}_{c,j} - \mathbf{P}_{S_{j'}}\left(l_c \beta \mathbf{v}_c - \sum_{j<l_c} \tilde{\mathbf{a}}_{c,j}\right)\right\|^2 + \sum_{i>c} \sum_{j=1}^{l_i} \|\tilde{\mathbf{a}}_{i,j}\|^2$$

$$= l_c^2 \beta^2 \|\mathbf{v}_c - \mathbf{P}_{S_{j'}} \mathbf{v}_c\|^2 + \phi_c$$

$$= l_c(\mathrm{OPT}_{k_c} - \phi_c)\|\mathbf{v}_c - \mathbf{P}_{S_{j'}} \mathbf{v}_c\|^2 + \phi_c$$

$$\geqslant l_c \mathrm{OPT}_{k_c} \|\mathbf{v}_c - \mathbf{P}_{S_{j'}} \mathbf{v}_c\|^2 - l_c \phi_c.$$

Note that $\lim_{\beta \to 0} \mathbf{P}_{S_{j'}} \mathbf{v}_c = \mathbf{0}$ because $\mathbf{v}_c$ is orthogonal to the subspace spanned by $S_{j'}$, so we can choose $\beta_c$ small enough so that $\|\mathbf{v} - \mathbf{P}_{S_{j'}} \mathbf{v}\|^2 \geqslant 1 - \frac{\delta}{2}$ for each $j' \in [l_c]$. Furthermore, we have

$$\phi_c = \sum_{i>c} \mathrm{tr}(\mathbf{B}_i) = \sum_{i>c} \alpha_i^2 l_i + \beta_i^2 l_i \leqslant 2\alpha_{c+1}^2 \sum_{i>c} l_i,$$

So, if we ensure that $\alpha_{c+1}^2 \leqslant \frac{\delta}{4} l_c \beta_c^2 / (\sum_{i>c} l_i)$, then:

$$l_c \phi_c \leqslant 2 l_c \alpha_{c+1}^2 \sum_{i>c} l_i \leqslant \frac{\delta}{2} \cdot l_c^2 \beta^2 \leqslant \frac{\delta}{2} l_c \cdot \mathrm{OPT}_{k_c},$$

which implies that $\mathrm{Er}_\mathbf{A}(S_{j'}) \geqslant (1-\delta) l_c \mathrm{OPT}_{k_c}$. Note that all the conditions we required on $\alpha_i$ and $\beta_i$ can be satisfied by a sufficiently quickly decreasing sequence $\alpha_1 \gg \beta_1 \gg \alpha_2 \gg \beta_2 \gg ... \gg \alpha_t \gg \beta_t > 0$, which completes the proof. $\qquad\square$

# G Proof of Corollary 1

**Corollary' 1.** *For $t \in \mathbb{N}$ and $\delta \in (0, 1)$, there is a sequence $k_1^l < k_1^u < k_2^l < k_2^u < \ldots < k_t^l < k_t^u$ and $\mathbf{A} \in \mathbb{R}^{m \times n}$ such that for any $i \in [t]$:*

$$\min_{S:|S|=k_i^l} \frac{\mathrm{Er}_{\mathbf{A}}(S)}{\mathrm{OPT}_{k_i^l}} \leqslant 1 + \delta \quad and$$

$$\min_{S:|S|=k_i^u} \frac{\mathrm{Er}_{\mathbf{A}}(S)}{\mathrm{OPT}_{k_i^u}} \geqslant (1 - \delta)(k_i^u + 1).$$

*Proof.* We will use Theorem 3 to construct the matrix $\mathbf{A}$ using the sequence we build below to make sure the upper and lower bounds are satisfied. Theorem 3 uses Lemma 4 to construct the matrix $\mathbf{A}$ which has a "step" eigenvalue profile i.e. there are multiple groups of eigenvalues and in each group the eigenvalue is constant (each group corresponds to a regular simplex, see Section 3). Below we consider a single such group that starts at $s = k_i^u$ and ends at $w = k_{i+1}^u$, and we let $k = k_{i+1}^l$, for any $i \in \{0, \ldots, t - 1\}$, with $k_0^u = 0$.

Theorem 1 implies that there is a set $S$ with an upper bound on the approximation factor $\mathrm{Er}_{\mathbf{A}}(S)/\mathrm{OPT}_k$ of $(1 + 2\epsilon)^2 \left(1 + \frac{s}{k-s}\right)\left(1 + \frac{k-s}{t_s-k}\right)$. Consider the following three conditions to ensure that each of the three terms in the above approximation factor is less than $(1 + \delta_1)$ where $\delta_1 = \delta/7$:

1. $\epsilon \leqslant \frac{(1+\delta_1)^{1/2}-1}{2} \implies (1 + 2\epsilon)^2 \leqslant (1 + \delta_1)$. Let $\tau_\epsilon = \frac{7}{\epsilon^4} \ln^2 \frac{1}{\epsilon}$, where $\epsilon$ is chosen so as to satisfy the above condition.

2. $k \geqslant \frac{s}{\delta_1} + s + \tau_\epsilon$ ensures that $(1 + \frac{s}{k-s}) \leqslant (1 + \delta_1)$ and that $k - s \geqslant \tau_\epsilon$.

3. $w \geqslant k(1 + \frac{1}{\delta_1}) + 1$.

To see the usefulness of condition 3, note that each group of vectors in column set of $\mathbf{A}$ constructed from Theorem 3 form a shifted regular simplex. A regular simplex has the smallest eigenvalue $0$ and the rest of the eigenvalues are all $(w - s)\alpha^2/(w - s - 1)$, where $\alpha$ is the length of each of the $(w - s)$ vectors in the simplex. Thus, we can lower bound the stable rank of the shifted simplex as $\mathrm{sr}_s(\mathbf{A}) \geqslant \frac{(w-s)\alpha^2}{(w-s)\alpha^2}(w - s - 1) = (w - s - 1)$. From condition 3:

$$w \geqslant k(1+\frac{1}{\delta_1})+1 \implies s + \mathrm{sr}_s(\mathbf{A}) \geqslant k(1+\frac{1}{\delta_1}) \implies t_s \geqslant k(1+\frac{1}{\delta_1})-\frac{s}{\delta_1} \implies 1+\frac{k-s}{t_s-k} \leqslant (1+\delta_1).$$

Thus if all the above three conditions are satisfied, the approximation ratio can be upper bounded by $(1 + \delta_1)^3 \leqslant (1 + \delta)$, since $\delta_1 = \delta/7$.

Similarly for the lower bound, we will need condition 4 below.

4. $w \geqslant \frac{2s}{\delta} + \frac{2}{\delta}$.

Now, we apply Theorem 3 using $k_i = w$ and $k_{i-1} = s$ to get the following lower bound with $\delta_2 = \delta/2$:

$$\min_{S:|S|=w} \frac{\mathrm{Er}_{\mathbf{A}}(S)}{\mathrm{OPT}_w} \geqslant (1 - \delta_2)(w - s) \geqslant (w + 1) - \frac{\delta}{2}(w + 1 + \frac{2s}{\delta} + \frac{2}{\delta}) \geqslant (1 - \delta)(w + 1),$$

where the last inequality follows from condition 4. Also, observe that we can replace conditions 3 and 4 with a single stronger condition: $w \geqslant k(1 + \frac{7}{\delta}) + 1 + \frac{2}{\delta}$.

We now iteratively construct the sequence that satisfies all of the above conditions:

1. $k_0^u = 0$

2. For $1 \leqslant i \leqslant t$

   (a) $k_i^l = \left\lceil \frac{7 k_{i-1}^u}{\delta} + k_{i-1}^u + \tau_\epsilon \right\rceil$.
   (b) $k_i^u = \lceil k_i^l (1 + 7/\delta) + \frac{2}{\delta} + 1 \rceil$.

We can now use Theorem 3 with subsequence $\{k_i^u\}$ which also constructs the matrix $\mathbf{A}$ through Lemma 4, to ensure that the lower bound of $(1 + \delta)(k_i^u + 1)$ is satisfied for $\mathbf{A}$ for all $i$. We can also use Theorem 1 for the same matrix $\mathbf{A}$ and $k = k_i^l$ for any $i$ to ensure that the upper bound of $(1 + \delta)$ is also satisfied for any $i$. $\qquad\square$

## H  Empirical evaluation with greedy subset selection

In this section, we provide a more detailed empirical evaluation to complement what we presented in Section 4. Our aim here is to demonstrate that our improved analysis of the CSSP/Nyström approximation factor can be useful in understanding the performance of not only the k-DPP method, but also of greedy subset selection. Note that our theory does not strictly apply to the greedy algorithm. Nevertheless, we show that, similar to the k-DPP method, greedy selection also exhibits the improved guarantees and the multiple-descent curve predicted by our analysis.

The most standard version of the greedy algorithm (see, e.g., Altschuler et al., 2016) starts with an empty set and then iteratively adds columns that minimize the approximation error at every step, until we reach a set of size $k$. The pseudo-code is given below.

---
**Greedy subset selection algorithm for CSSP/Nyström**

**Input:** $k \in [n]$    and    an $m \times n$ matrix $\mathbf{A}$ (CSSP),    or    an $n \times n$ p.s.d. matrix $\mathbf{K} = \mathbf{A}^\top \mathbf{A}$ (Nyström)

$S \leftarrow \varnothing$
**for** $i = 1$ **to** $k$ **do**
    Pick $i \in [n] \backslash S$ that minimizes $\mathrm{Er}_\mathbf{A}(S \cup \{i\})$,   or equivalently, $\|\mathbf{K} - \widehat{\mathbf{K}}(S \cup \{i\})\|_*$
    $S \leftarrow S \cup \{i\}$
**end for**

**return** $S$

---

In our empirical evaluation we use the same experimental setup as in Section 4, by running greedy on a toy dataset with the linear kernel $\langle \mathbf{a}_i, \mathbf{a}_j \rangle_\mathrm{K} = \mathbf{a}_i^\top \mathbf{a}_j$ that has one sharp spectrum drop (controlled by the condition number $\kappa$), and two Libsvm datasets with the RBF kernel $\langle \mathbf{a}_i, \mathbf{a}_j \rangle_\mathrm{K} = \exp(-\|\mathbf{a}_i - \mathbf{a}_j\|^2/\sigma^2)$ for three values of the RBF parameter $\sigma$. The main question motivating these experiments is: does the approximation factor of the greedy algorithm exhibit the multiple-descent curve that is predicted in our analysis, and are the peaks in this curve aligned with the sharp drops in the spectrum of the data?

The plots in Figure 4 confirm that the Nyström approximation factor of greedy subset selection exhibits similar peaks and valleys as those indicated by our theoretical and empirical analysis of the k-DPP method. This is most clearly observed for the toy dataset (Figure 4 left), where the peak grows with the condition number $\kappa$, and for the *bodyfat* dataset (Figure 4 center), where the size of the peak is proportional to the RBF parameter $\sigma$. Moreover, we observe that when the spectral decay is slow/smooth, which corresponds to smaller values of $\sigma$, then the approximation factor of the greedy algorithm stays relatively close to 1. For the *eunite2001* dataset (Figure 4 right), the behavior of the approximation factor is very non-linear, with several peaks occurring for large values of $\sigma$. Interestingly, while the peaks do align with some of the drops in the spectrum, not all of the spectrum drops result in a peak for the greedy algorithm. This goes in line with our analysis, in the sense that a sharp drop in the spectrum following the $k$th eigenvalue is a *necessary but not sufficient* condition for the approximation factor of the optimal subset $S$ of size $k$ to exhibit a peak.

Our empirical evaluation leads to an overall conclusion that the multiple-descent curve of the CSSP/Nyström approximation factor is a phenomenon exhibited by both *randomized* methods, such as the k-DPP, and *deterministic* algorithms, such as greedy subset selection. While the exact behavior of this curve is algorithm-dependent, significant insight can be gained about it by studying the

Figure 4: Top plots show the Nyström approximation factor $\|\mathbf{K} - \widehat{\mathbf{K}}(S)\|_* / \mathrm{OPT}_k$, where $S$ is constructed using greedy subset selection, against the subset size $k$, for a toy dataset ($\kappa$ is the condition number) and two Libsvm datasets ($\sigma$ is the RBF parameter). Bottom plots show the spectral decay for the top $40$ eigenvalues of each kernel $\mathbf{K}$, demonstrating how the peaks in the Nyström approximation factor align with the drops in the spectrum.

spectral properties of the data. Our results suggest that performing a theoretical analysis of the multiple-descent phenomenon for greedy methods is a promising direction for future work.