[Reviews · NeurIPS 2020]

Review 1

Summary and Contributions: The paper studies the important column subset selection problem (as well as extensions to the Nyström method), where the aim is to find a subset of k columns such that the projection cost of all points to the span of selected points is minimized. Classic randomized algorithms for the problem could only guarantee a worst-case approximation factor of O(k+1), and a lower bound construction was known where the approximation is \Omega(k). It is also known that the algorithms often perform much better than this worst case. The paper thus inspects the behavior of upper and lower bounds parameterized by k in more detail by providing a kind of "beyond worst case analysis", that still considers the worst case but for each value of k (or regions thereof) individually rather than only for the worst case choice of k. The main contributions are: 1. A sequence of upper bounds (for sampling the columns via a k-Determinantal Point Process (k-DPP)), depending on a stable rank parameter, that coincide with the Theta(k) bound at values of k where the spectrum has significant jumps. But the new bounds have significantly lower "valleys" in the regions between those jumps. E.g. for all values of k lower than the stable rank (of order 0) a simple consequence of their "master theorem" is a O(sqrt(k)) upper bound. They call this behavior with multiple peaks and valleys "multiple descent curve". 2. An application to matrices where the spectrum does't possess sudden jumps but a "regular" decay. For polynomial decay (of degree p) the bound becomes O(p). For an exponential decay, (1-delta)^i the bound becomes (1+delta k) which is better than the worst case whenever delta is o(1). 3. A worst case construction where the spectrum has a sequence of jumps, so that the lower bound becomes large at those jumps but the error remains upper bounded by a constant between those peaks. This is done by putting the known single jump construction into orthogonal subspaces so they have independent impact. This construction shows that the "multiple descent curve" structure of the upper and lower bounds is inherent to the problem, rather than an artifact of the k-DPP sampling method or of the analysis. (additionally the behavior is also observed for a greedy selection algorithm in the supplement) 4. It is explained how the above results extend to the Nyström approximation with trace norm error. This also yields further upper bounds in terms of defining parameters of the RBF, and Matern kernels. ######################## Dear authors, thanks for your clarifying response to my questions. This enforces my decision to keep my initial high rating. Two final comments: - Indeed, Roughgarden [2019] would be a good reference and quite general survey on beyond-worst-case analysis. Maybe you can find one or two more examples among the recent top notch AI/ML publications. - Correctness of sr(A)-1 < k < sr(A): I think at least the upper inequality should be "lower equal" (sr(A)-1 < k <= sr(A)) to allow integer values of sr(A) and k. Please double check again. Thanks

Strengths: The paper makes significant progress in the study of the CSSP and Nyström approximation problems. Notably under the original constraint of subsets with at most k elements. (no oversampling allowed) It provides a natural form of beyond worst case analysis with highly non-trivial upper and lower bounds dependent on the subset size. It provides experimental assessments of the theoretical claims that even further support the intuition.

Weaknesses: no considerable weaknesses

Correctness: all claims seem correct; the empirical methodology satisfies highest scientific standards clearly stating the research question, explaining how it is assessed and answering the questions by drawing correct conclusions from the experiment.

Clarity: The paper is very well written. It is one of few papers that have the perfect balance between formal technical and intuitive explanations in the main writeup, and additional much more detailed technical results in the supplement.

Relation to Prior Work: There is an extensive discussion on related work (partly in the supplement). It is clearly stated what was known before and what are the new contributions in the present work. I was missing only one thing: there should be some references on the area of "beyond worst case analysis". Maybe you could add a survey on this topic and one or two examples of beyond worst case analysis among recent ICML or NeurIPS papers.

Reproducibility: Yes

Additional Feedback: some minor comments to improve the paper: - add references for "beyond worst-case analysis" - 89: is sr(A)-1 < k < sr(A) correct? if sr(A) is an integer, there is no value of k to satisfy this. maybe the upper bound should be rank(A) instead of sr(A) ? - 92: O(sqrt(k)) to O(k) -> "O(sqrt(k)) to \Omega(k)" -111: "k between 20 and 50" -> "k between 20 and 40" - 169: explain parameters \nu and \ell. - 171: note that Matern turns to RBF with \ell = \sigma the limit for nu -> \infty. Also note that this limit is approached very quickly so that in practice \nu is almost always bounded by a small constant 7/2 (Rasmussen & Williams, 2006), which supports even more that O(1+nu) is indeed a very good approximation factor! - "sufficiently wide valley". There is a quantification of "sufficiently wide" in the lemmas. Maybe it would be nice to have it also somewhere in the text, especially since the expression can become quite a large "constant" depending on eps. - Figure 2: (top) -> (left), and (bottom) -> (right) - Figure 2: is the min Phi_s(k) line taken over s\in{10, 15, 25} or over all values of s\in [0,rank(A)] ? - In the proof of thm 3 I am missing one additional argument concluding wrt the "key challenge" described above lemma 4. What is the consequence of the simplices being orthogonal to one another?


Review 2

Summary and Contributions: This paper studies Column Subset Selection (CSS) problem. They proposed a new algorithm of which approximation ratio depends on the stable rank of the input matrix. They also provided the lower bound. Their upper and lower bounds show that for some matrices the CSS approximation factor can exhibit peaks and valleys as a function of the subset size k。

Strengths: The algorithm is novel. It is surprising that running Determinantal point processes (DPP) on a proper scaled matrix can give a good CSS with approximation factor depending on the stable rank. It is very clean and easy to implement. The construction of the hard instance is interesting. The combination of the upper bound and the lower bound reveals an interesting phenomenon that the optimal approximation ratio may be a multiple-descent curve.

Weaknesses: For the upper bound side, the main issue is that the analysis breaks for small k. It is not clear the behavior of the algorithm when k is a small constant.

Correctness: I did not find any explicit issue in the proofs or experiments.

Clarity: The paper is clean and well written in general.

Relation to Prior Work: In Appendix, authors gave detailed comparison with previous work.

Reproducibility: Yes

Additional Feedback: - line 47: should mention the definition of DPP is in appendix. - I felt that the authors missed some related work. For example, "Algorithms for $\ell_p$ Low-Rank Approximation" (Chierichetti et. al., ICML'2017) gives a iterative column selection methods. An another algorithm "Optimal CUR Matrix Decompositions" (Boutsidis & Woodruff, STOC'2014) uses leverage score and adaptive sampling methods. It would be good that authors can discuss whether these algorithms can achieve the similar guarantees as the proposed algorithm or not. Post rebuttal: Authors' response looks good to me. I tend to accept the paper.


Review 3

Summary and Contributions: The paper considers Column Subset Selection Problem (CSSP) (a variant also known as the Nystro¨m method) which selects a subset S of k columns from an m by n matrix, so that the difference bewteen the matrix and its projection on the span of S is small (in terms of Frobenius norm). It focuses on the ratio between this error and that of the best rank k approximation. It exploits the spectral properties of the matrix and gets better bounds than known worst-case bounds for matrices with good singular value decay. It also shows that the approximation ratio as a function of k has multiple peaks and valleys, and shows that this is inherent by providing a lower bound. Finally, it empirically verifies the phenomenon using the RBF kernel on real datasets. pros: + The paper explains why the method works well in practice, much better than predicted by the worst-case analysis. It shows that the method makes good use of the spectral decay of the data, which is presented in many practical datasets. + It further observes the interesting multiple descent phenomenon of the approximation ratio for some matrices (i.e., with multiple significant drops in the spectral decay). Importantly, it shows that this is not an artifact of their analysis but rather an inherent property of the task on such matrices, by showing a lower bound that, together with the upper bound, justifies the phenomenon. cons: - The connection of multiple-descent phenomenon with the spectral phase transition has been observed in the previous works, as discussed in the paragraph of Line 128. While differences and similarities are mentioned, one commonality not discussed is that a bit overparameterization can help the learning significantly. Maybe the authors can comment more on whether this help of overparameterization is due to the same/similar reasons as in previous works.

Strengths: + The paper explains why the method works well in practice, much better than predicted by the worst-case analysis. It shows that the method makes good use of the spectral decay of the data, which is presented in many practical datasets. + It further observes the interesting multiple descent phenomenon of the approximation ratio for some matrices (i.e., with multiple significant drops in the spectral decay). Importantly, it shows that this is not an artifact of their analysis but rather an inherent property of the task on such matrices, by showing a lower bound that, together with the upper bound, justifies the phenomenon.

Weaknesses: - The connection of multiple-descent phenomenon with the spectral phase transition has been observed in the previous works, as discussed in the paragraph of Line 128. While differences and similarities are mentioned, one commonality not discussed is that a bit overparameterization can help the learning significantly. Maybe the authors can comment more on whether this help of overparameterization is due to the same/similar reasons as in previous works. ===============after response=============================== The authors haven't given an adequate answer to my question on connection to more general overparameterization in machine learning, but that's an question beyond the scope of the paper anyway, and so my score remains the same.

Correctness: yes

Clarity: Yes

Relation to Prior Work: Yes. I will apprecitate more discussion about the effect of overparameterization in the task.

Reproducibility: Yes

Additional Feedback:


Review 4

Summary and Contributions: This is a high-quality theoretical paper that provides unique insights on the origins of the multiple descent curve for feature selection. The master theorem as function of the spectral decay and the lower bound helps us to understand why we can observe multiple peaks and descents of the approximation factor. The supplementary material is also very useful providing excellent context and additional experiments.

Strengths: High quality rigorous paper that provides upper and lower bounds that helps us understand the qualitative and quantitative of the expected approximation factor.

Weaknesses: Although the paper is mostly theoretical I wish the authors could have shown the analysis on other data sets with different types of spectral decays. Having such an empirical analysis can educate us in what to expect in different situations. If we compare Fig. 3 and the supplementary material for k-DPP and the greedy method only the enuite2001 spectral decay appears to have a differing approximation factor. Why? The smaller eigenvalues are typically quite noisy and just by sorting then they may appear as having an exponential spectral decay. Yet, they are likely reflecting the sampling of an exponential distribution rather than the actual decay. Have the authors thought how to discard/correct for it?

Correctness: I did not find obvious errors in the manuscript.

Clarity: very clearly written.

Relation to Prior Work: properly discussed.

Reproducibility: Yes

Additional Feedback: I enjoyed learning about the relationship of the spectral decay and the approximation factor. Thank you.

[Author Response · NeurIPS 2020]

We thank all the reviewers for the positive comments.

**R1: Correctness of sr(A)-1 < k < sr(A)**  Yes, this statement is correct. Our goal in Remark 1 is to illustrate a sharp
transition in the approximation factor as $k$ approaches the stable rank from below. Remark 1(2) is an existential result
i.e. there exists an $\mathbf{A}$ for which the lower bound on the approximation factor holds.

**R1: L92 "stable rank of A is 20" should be 40**  Here, we are referring to $sr_0(\mathbf{A})$, which is 20 and in Figure 1 it
coincides with the first peak in the approximation factor.

**R1: Minor typos/comments**  Thanks for the careful read. We will make the suggested changes in the final version.

**R1: Fig 2. Is the min $\Phi_s(k)$ line taken over $s \in \{10, 15, 25\}$ or over all values of $s \in [0, \mathbf{rank}(\mathbf{A})]$**  The minimum
is taken over all values of $s$.

**R1: Proof of Thm 3. Importance of orthogonality of simplices**  For our setup, because of multiple simplices, the
symmetry that holds for Deshpande et al. [2006] breaks, and thus different subsets of the same size have different
possible errors. Orthogonality, along with the choice of $\alpha_i, \beta_i$, ensures that we can pinpoint the best possible set of size
$k$ and its corresponding error amongst the different choices (see proof of Lemma 4 in Appendix F).

**R1: Other works on beyond worst case analysis**  We feel that any thorough discussion of works on "beyond worst-
case analysis" would have to span a very wide range of topics, which is beyond the scope of this paper. In the final
version, we will reference Roughgarden [2019] which provides a broader discussion on the area. If the reviewer has any
other works in mind which are particularly relevant, we will definitely consider adding them.

**R2: Suggested additional related works on matrix approximation**  The list of published papers on algorithms for
matrix approximation is indeed extensive. We thank the reviewer for the additional pointers, as they are relevant to
our work. In particular, we believe that extending our results to the $\ell_p$ low-rank approximation setting is an interesting
direction for future work.

**R3: Connections with overparametrization**  While we are not aware of a formal connection between the multiple-
descent phenomenon in column subset selection and overparameterization in machine learning, both settings have to do
with obtaining a stable approximation of the data, and this stability can often be captured by the spectral properties of
the problem.

**R4: Difference between k-DPP and greedy on enuite2001**  We also found the different behaviors of the approxi-
mation factor for the two methods intriguing. One explanation for this is that the expected approximation factor for
a k-DPP is fully determined by the eigenvalues, whereas the performance of the greedy method depends on other
characteristics of the data. However, it is not clear to us why this is most pronounced on the eunite2001 dataset.

**R4: Correcting the effect of noisy eigenvalues**  We have not considered this, but it is indeed an interesting question.
One simple observation is that if one chooses the subset size $k$ small enough so that the noisy tail of the spectrum
consists of only the eigenvalues with index sufficiently larger than $k$, then the effect of noise on the approximation
factor should be minimal.

# References

A. Deshpande, L. Rademacher, S. Vempala, and G. Wang. Matrix approximation and projective clustering via volume
sampling. In *Proceedings of the Seventeenth Annual ACM-SIAM Symposium on Discrete Algorithm*, pages 1117–1126,
Miami, FL, USA, January 2006.
T. Roughgarden. Beyond worst-case analysis. *Communications of the ACM*, 62(3):88–96, 2019.


[Meta-Review · NeurIPS 2020]

The paper gives improved approximation guarantees for the column subset selection problem (CSSP) and the Nystrom’s method for low-rank approximation of large datasets. The analysis reveals a multiple descent phenomenon which authors argue is real and not an artifact of their analysis. The paper is written clearly, and the results and insights in the paper are compelling. Overall, a good paper. Accept!